# The AlborEX dataset: sampling of submesoscale features in the Alboran Sea

Charles Troupin[1], Ananda Pascual[2], Simon Ruiz[2], Antonio Olita[3], Benjamin Casas[2], Félix Margirier[4], Pierre-Marie Poulain[5], Giulio Notarstefano[5], Marc Torner[6], Juan Gabriel Fernández[6], Miquel Àngel Rújula[6], Cristian Muñoz[6], Eva Alou[6], Inmaculada Ruiz[6], Antonio Tovar-Sánchez[7], John T. Allen[6], Amala Mahadevan[8], and Joaquín Tintoré[6,2]

[1]GeoHydrodynamics and Environment Research (GHER), Freshwater and OCeanic science Unit of reSearch (FOCUS), University of Liège, Liège, Belgium
[2]Instituto Mediterráneo de Estudios Avanzados (IMEDEA, CSIC-UIB), Esporles, Spain
[3]Institute for Coastal Marine Environment-National Research Council (IAMC-CNR) Oristano, Oristano, Italy
[4]Sorbonne Universités (UPMC, Univ Paris06)-CNRS-IRD-MNHN, Laboratoire LOCEAN, Paris, France
[5]Istituto Nazionale di Oceanografia e di Geofisica Sperimentale (OGS), Trieste, Italy
[6]Balearic Islands Coastal Observing and Forecasting System (SOCIB), Palma de Mallorca, Spain
[7]Instituto de Ciencias Marinas de Andalucía, (ICMAN – CSIC), Puerto Real, Spain
[8]Woods Hole Oceanographic Institution, Woods Hole, MA, USA

**Correspondence:** Charles Troupin (ctroupin@uliege.be)

**Abstract.** AlborEX (Alboran Sea Experiment) consisted of a multi-platform, multi-disciplinary experiment carried out in the Alboran Sea (Western Mediterranean Sea) between May 25 and 31, 2014. The observational component of AlborEx aimed to sample the physical and biogeochemical properties of oceanographic features present along an intense frontal zone, with a particular interest in the vertical motions in its vicinity. To this end, the mission included 1 research vessel (66 profiles), 2 underwater gliders (adding up 552 profiles), 3 profiling floats and 25 surface drifters.

Near real-time ADCP velocities were collected nightly and during the CTD sections. All of the profiling floats acquired temperature and conductivity profiles, while the Provor-bio float also measured oxygen and chlorophyll-a concentrations, colored dissolved organic matter, backscattering at 700 nm, downwelling irradiance at 380, 410, 490 nm, and photo-synthetically active radiation (PAR).

In the context of mesoscale and submesoscale interactions, the AlborEX dataset constitutes a particularly valuable source of information to infer mechanisms, evaluate vertical transport and establish relationships between the thermal and haline structures and the biogeochemical variable evolution, in a region characterised by strong horizontal gradients provoked by the confluence of Atlantic and Mediterranean Waters, thanks to its multi-platform, multi-disciplinary nature.

The dataset presented in this paper can be used for the validation of high-resolution numerical models or for data assimilation experiment, thanks to the various scales of processes sampled during the cruise. All the data files that make up the dataset are available in the SOCIB data catalog at https://doi.org/10.25704/z5y2-qpye. The nutrient concentrations are available at https://repository.socib.es:8643/repository/entry/show?entryid=07ebf505-bd27-4ae5-aa43-c4d1c85dd500.

# 1 Introduction

The variety of physical and biological processes occurring in the ocean at different spatial and temporal scales requires a combination of observing and modelling tools in order to properly understand the underlying mechanisms. Hydrodynamical models make it possible to design specific numerical experiments or simulate idealised situation that can reproduce some of these processes and assess the impacts of climate change. Despite the continuous progresses made in modeling (spatial resolution, parameterization, atmospheric coupling, ...), in situ observations remain an essential yet challenging ingredient when addressing the complexity of the ocean.

To properly capture and understand these small-scale features, one cannot settle for only observations of temperature and salinity profiles acquired at different times and positions, but rather has to combine the information from diverse sensors and platforms acquiring data at different scales and at the same time, similarly to the approach described in Delaney and Barga (2009). This also follows the recommendation for the Marine Observatory in Crise et al. (2018), especially the co-localization and synopticity of observations and the multi-platform, adaptive sampling strategy. We will refer to this as multi-platform systems, by opposition to experiments articulated only around the observations made using a research vessel. Further details can be found in Tintoré et al. (2013).

The western Mediterranean Sea is a particularly relevant region for multi-platform experiments, thanks to the wide range of processes taking place and intensively studied since the work of Wüst (1961) on the vertical circulation: influence on climate (e.g., Giorgi, 2006; Giorgi and Lionello, 2008; Adloff et al., 2015; Guiot and Cramer, 2016; Rahmstorf, 1998) and sea-level change (e.g., Tsimplis and Rixen, 2002; Bonaduce et al., 2016; Wolff et al., 2018), thermohaline circulation (e.g., Bergamasco and Malanotte-Rizzoli, 2010; Millot, 1987, 1991, 1999; Skliris, 2014; Robinson et al., 2001), water mass formation and convection process (e.g., MEDOC-Group, 1970; Stommel, 1972; Send et al., 1999; Macias et al., 2018), mesoscale (e.g., Alvarez et al., 1996; Pinot et al., 1995; Pujol and Larnicol, 2005; Sánchez-Román et al., 2017) and submesoscale processes (e.g., Bosse et al., 2015; Damien et al., 2017; Margirier et al., 2017; Testor and Gascard, 2003; Testor et al., 2018). Other recent instances of multi-platform experiments in the Mediterranean Sea were focused on the Northern Current (December 2011, Berta et al., 2018), deep convection in the Northwestern Mediterranean sea (July 2012–October 2013, Testor et al., 2018), the Balearic Current system (July and November 2007, April and June 2008, Bouffard et al., 2010) and coastal current off west of Ibiza island (August 2013, Troupin et al., 2015). Similar studies comparing almost synchronous glider and SARAL/AltiKa altimetric data on selected tracks have also been carried between the Balearic Islands and the Algerian coasts (Aulicino et al., 2018; Cotroneo et al., 2016).

Recently, the efforts carried out by data providers and oceanographic data centers through European initiatives such as SeaDataNet (http://seadatanet.org/) makes possible the creation and publication of aggregated datasets covering different European regional seas, including the Mediterranean Sea (Simoncelli et al., 2014), upon which hydrographical atlas are build (e.g. Simoncelli et al., 2016; Iona et al., 2018b). These atlas are particularly useful for the description of the general circulation, the large-scale oceanographic features or for the assessment of the long-term variability (Iona et al., 2018a). However their limita-

tion to temperature and salinity variables (as of July 2018) and their characteristic spatial scale prevent them to be employed for the study of submesoscale features.

The AlborEx multi-platform experiment was performed in the Alboran Sea from from May 25 to 31, 2014, with the objective of capturing meso and submesoscale processes and evaluating the interactions between both scales, with a specific focus on the vertical velocities. The observing system, described in the next section, is made up of the SOCIB coastal R/V, 2 underwater gliders, 3 profiling floats and 25 surface drifters, complemented by remote-sensing data (sea surface temperature and chlorophyll concentration). The resulting data set is particularly rich thanks to the variety of sensors and measured variables concentrated on a relatively small area.

Section 2 strives to summarize the motivations behind the sampling and deployments. The presentation of the available data is the object of the Section 4.

## 2 The AlborEx mission

The mission took place from May 25 to May 31, 2014 in the Alboran Sea frontal system (Cheney, 1978; Tintoré et al., 1991, see Fig. 1), scene of the confluence of Atlantic and Mediterranean waters. The mission itself is extensively presented in Ruiz et al. (2015) and the features and processes captured by the observations are discussed in Pascual et al. (2017). Olita et al. (2017) examined the deep chlorophyll maximum variation combining the bio-physical data from the gliders and the profiling floats. The present papers focuses solely on the description of the original dataset, graphically summarised in Fig. 1).

### 2.1 General oceanographic context

The definitive sampling area was not firmly decided until a few days before the start of the mission. Prior to the experiment, satellite images of sea surface temperature (SST) and chlorophyll-a concentration were acquired from the Ocean Color Data server (https://oceandata.sci.gsfc.nasa.gov/, last accessed August 3, 2018) in order to provide an overview of the surface oceanic features apparent in the Alboran Sea. A well-defined front separating Atlantic and Mediterranean waters and exhibiting filament-like structures was selected as the study area (see rectangular boxes in Figs. 1 and 2).

The pair of images indicates that the front position slightly changed between May 25 and 30. An anticyclonic eddy centered around 36°30'N, 0°30'W, according to altimetry data (not shown), slowly followed an eastward trajectory in the following days. Other SST images during the period of interest (not shown here) displayed different temperature values near the front, yet the front position remains stable.

### 2.2 Design of the experiment

The deployment of in situ systems was based on the remote-sensing observations described in the previous Section. Two high-resolution grids were sampled with the research vessel, covering an approximative region of 40 km × 40 km. At each station, one CTD cast and water samples for chlorophyll concentrations and nutrients analysis were collected. The thermosalinograph observations were also used in order to assess the front position.

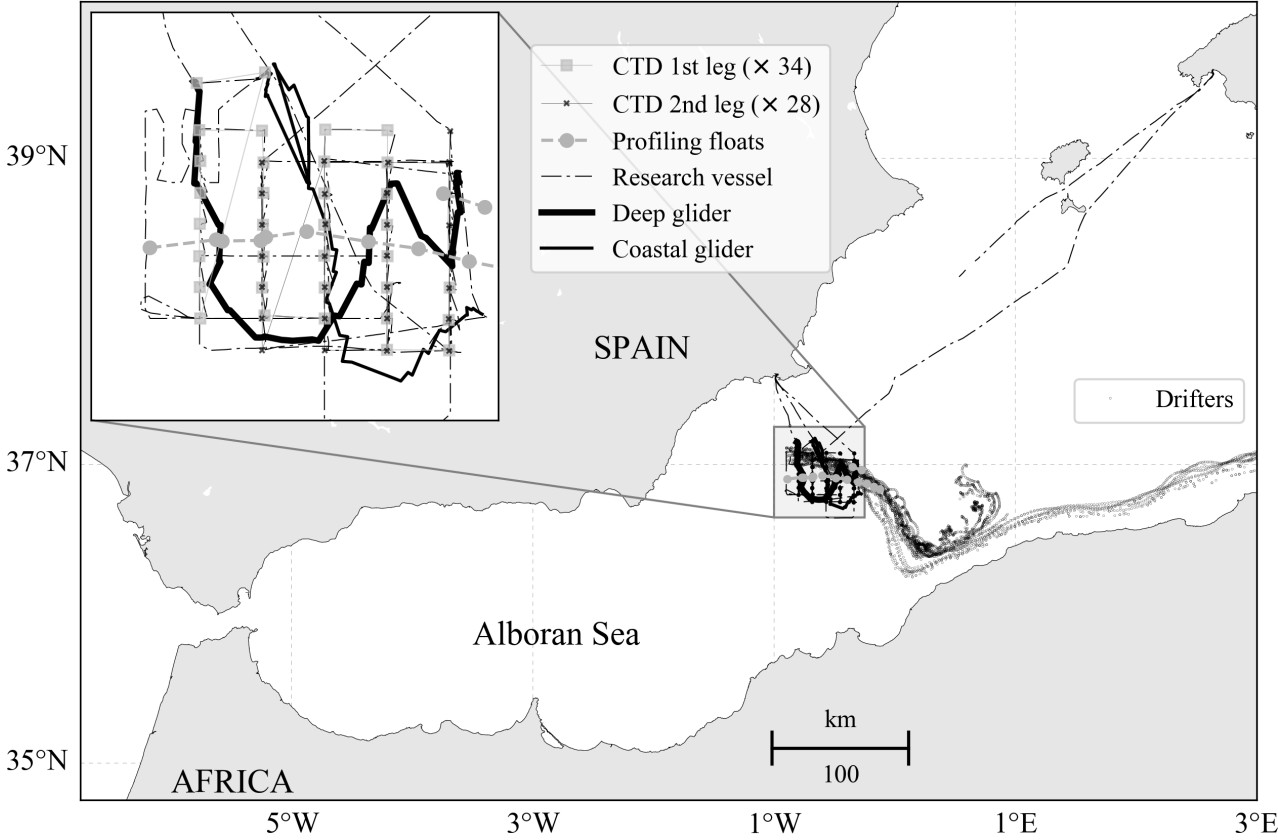

**Figure 1.** Area of study, positions and trajectories of the main platforms. The close-up view on displays the glider and the CTD measurements.

One deep glider and one coastal glider were deployed in the same area with the idea to have butterfly-like track across the front. These idealised trajectories turned out to be impossible considering the strong currents occurring in the region of interest at the time of the mission.

5    The 25 drifters were released close to the frontal area with the objective to detect convergence and divergence zones. Their release locations were separated by a few kilometers.

### 2.3   Data reuse

Three main types of data reuse are foreseen: 1. model validation, 2. data assimilation (DA) and 3. planning of similar in situ experiments.

10    With the increase of spatial resolution in operational models, the validation at the smaller scales requires high-resolution observations. Remote-sensing measurements such as SST or chlorophyll-a concentration provides a valuable source of information but are limited to the surface layer. In the case of the present experiment, the position, intensity (gradients) and vertical

structure of the front represent challenging features for numerical models, even when data assimilation is applied (Hernandez-Lasheras and Mourre, 2018).

The AlborEx dataset can be used for DA experiments, for example assimilating the CTD measurements in the model and using the glider measurements as an independent observation dataset. The assimilation of glider observations has already been performed in different regions (e.g. Melet et al., 2012; Mourre and Chiggiato, 2014; Pan et al., 2014) and has been shown to improve the forecast skills. However the assimilation of high-resolution data is not trivial: the the background error covariances tends to smooth the small scale features present in the observations and the high density of measurements may require the use of super-observations (averaging the observations in the model cells). Another complication arises from the fact that the observational errors are correlated, while data assimilation schemes often assume those errors are not correlated.

Finally, other observing and modeling programs in the Mediterranean Sea can also benefit from the present dataset, for instance the Coherent Lagrangian Pathways from the Surface Ocean to Interior (CALYPSO) in the Southwest Mediterranean Sea (Johnston et al., 2018). Similarly to AlborEx, CALYPSO strives to study a strong ocean front front and the vertical exchanges taking place in the area of interest. For details on the mission objectives, see https://www.onr.navy.mil/Science-Technology/Departments/Code-32/All-Programs/Atmosphere-Research-322/Physical-Oceanography/CALYPSO-DRI, last accessed December 17, 2018.

## 2.4   Processing levels

For each of the platform described in Sec. 2, different processing are performed with the objective to turn raw data into quality-controlled, standardised data directly usable by scientists and experts. Specific conventions for data managed by SOCIB are explained below.

All the data provided by SOCIB are available in different, so-called processing levels, ranging from 0 (raw data) to 2 (gridded data). The files are organized by *deployments*, a deployment being defined as an event initiated when an instrument is put at sea and finished once the instrument is recovered from sea. Table 1 summarizes the deployments performed during the experiment and the available processing levels.

**Level 0 (L0)** : this is the level closest to the original measurements, as it is designed to contain exactly the same data as the raw files provided by the instruments. The goal is to deliver a single, standardised netCDF file, instead of one or several files in a platform-dependent format.

**Level 1 (L1)** : in this level, additional variables are derived from the existing ones (e.g., salinity, potential temperature). The attributes corresponding to each variable are stored in the netCDF file, with details of any modifications. Unit conversion are also applied if necessary.

**Level 1 corrected (L1_corr)** : this level is only available for the CTD: a corrective factor is obtained by a linear regression between the salinity measured by the CTD and that measured by the salinometer. The files corresponding to that processing levels contain new variables of conductivity and salinity to which the correction was applied. Additional metadata regarding the correction are also provided in the file.

**Level 2 (L2)** : this level is only available for the gliders. It consists of regular, homogeneous and instantaneous profiles obtained by gridding the L1 data. In other words, 3-dimensional trajectories are transformed into a set of instantaneous, homogeneous, regular profiles. For the spatial and temporal coordinates: the new coordinates of the profiles are computed as the mean values of the cast readings. For the variables: a binning is performed, taking the mean values of readings in depth intervals centered at selected depth levels. By default, the vertical resolution (or bin size) is set to 1 meter. This level was created mostly for visualization purposes.

The glider data require a specific processing to ingest and convert the raw data files produced by the coastal and deep units. This is done within a toolbox designed for this purpose and extensively described in Troupin et al. (2016), the capabilities of which includes metadata aggregation, data download, advanced data processing and the generation of data products and figures. Of particular interest is the application of a thermal-lag correction for un-pumped Sea-Bird CTD sensors (Garau et al., 2011), which improves the quality of the glider data.

**Table 1.** Characteristics of the instrument deployments in AlborEx.

| Instruments | Number of deployments | Initial time | Final time | Processing levels | | |
|---|---|---|---|---|---|---|
| | | | | L0 | L1 | L2 |
| Weather station on board R/V | 1 | 2014-05-25 | 2014-05-02 | ✓ | ✓ | |
| ADCP on board R/V | 1 | 2014-05-25 | 2014-05-02 | ✓ | ✓ | |
| CTD | 1 (66 stations) | 2014-05-25 | 2014-05-02 | ✓ | ✓ | |
| Gliders | 2 | 2014-05-25 | 2014-05-30 | ✓ | ✓ | ✓ |
| Surface drifters | 25 | 2014-05-25 | beyond the experiment | ✓ | ✓ | |
| Profiling floats | 3 | 2014-05-25 | beyond the experiment | ✓ | ✓ | |

## 2.5 Quality control

Automated data QC is part of the processing routine of SOCIB Data Center: most of the datasets provided with this paper come with a set of flags that reflect the quality of the measurements, based on different tests regarded the range of measurements, the presence of spike, the displacement of the platform and the correctness of the metadata.

The QC are based on existing standards for most of the platforms. They are extensively described in the Quality Information Document (SOCIB Data Center, 2018). The description platform by platform is provided in the next Section.

### 2.5.1 Quality flags

The flags used on the data are described in Tab. 2.

**Table 2.** Quality Control Flags.

| Code | Meaning |
| --- | --- |
| 0 | No QC was performed |
| 1 | Good data |
| 2 | Probably good data |
| 3 | Probably bad data |
| 4 | Bad data |
| 6 | Spike |
| 8 | Interpolated data |
| 9 | Missing data |

### 2.5.2 QC tests

The main tests performed on the data are:

**range:** depending on the variable considered, low and high threshold are assigned. First there is a global range: if the measured values falls outside, then the flag is set to 4 (bad data). Then a regional range test is applied: the measurements outside this range are assigned the flag 2 (probably good).

**spike:** the test consists in checking the difference between sequential measurements (i.e. not measured at the same time). For the $\jmath$-th measurement:

$$\text{spike} = \left| V_{\jmath} - \frac{V_{\jmath+1} + V_{\jmath-1}}{2} \right| - \left| \frac{V_{\jmath+1} - V_{\jmath-1}}{2} \right|$$

When the spike value is above the threshold (depending on the variable), the flag is set to 6.

**gradient:** it is computed for the variables along different coordinates (horizontal, depth, time).

**stationarity:** it aims to checks if measurements exhibit some variability over a period of time, by computing the difference between the extremal values over that period.

It is worth mentioning the tests described above are not yet applied on the glider data, since their processing is done outside of the general SOCIB processing chain, but the tests have been implemented in the glider toolbox (Troupin et al., 2016, and available at https://github.com/socib/glider_toolbox) and will be made operational once they have been properly tested and validated.

As the new files will not be available before a full reprocessing of all the historical missions, the decision was taken to provide the data files in their current state. A new version will be uploaded as soon as the processing has been performed.

# 3 In situ observation

Whereas the remote sensing measurements helped in the mission design and the front detection, in situ observation were essential to fulfill the mission objectives. The different platforms deployed for the data collection are presented hereinafter.

## 3.1 Research vessel

The SOCIB coastal research vessel (R/V) was used to sample the area with vertical profiles acquired though the CTD. Two distinct CTD legs were performed on a 10 km × 5 km resolution grid, as depicted in Fig. 3: the first survey was run from May 26 to 27 and consisted of 34 casts along 5 meridional legs. The second survey took place from May 29 to 30 and was made up of 28 casts. The casts from both surveys were performed at almost similar locations in order to allow for detecting changes between the two periods. On average the profiles reached a maximal depth of approximatively 600 m.

The distinct water properties on both sides of the front are evidenced by the T-S diagrams in Fig. 4, where the colors represent the fluorescence. The salinity range north of the front is roughly between 38 and 38.5, with the exception of a few measurements, and confirms the nature of the Mediterranean Water mass. The fluorescence maximum appears between 14 and 15°C. South of the front the salinity range is wider while the temperature values are similar to the north.

In addition to the CTDs, the R/V thermosalinograph continuously acquired temperature and conductivity along the ship track, from which near surface salinity is derived (Fig. 5). The R/V weather station acquired air temperature, pressure, wind speed and direction during the whole duration of the mission. Direct measurements of currents were performed with acoustic Doppler current profiler and are presented in Sec. 3.5.

### 3.1.1 Configuration

The CTD rosette was equipped with:

- a Sea-Bird SBE 911Plus, 2 conductivity and temperature sensors and 1 pressure sensor units,

- a SBE 43 oxygen sensor,

- a Seapoint [FTU] fluorescence and turbidity sensor.

The GEONICA METEODATA 2000 weather station measured the following variables: air pressure, temperature, humidity, wind speed and direction, with a resolution of 10 minutes. The continuous, near-surface measurements of temperature and salinity are provided by a SeaBird SBE21 thermosalinograph.

### 3.1.2 Quality control

The general checks described in Sec. 2.5.2 (i.e., ranges, spike, gradient and stationarity) are applied on the temperature, salinity, conductivity and turbidity. The threshold values are detailed in the corresponding tables in the QC procedure document (SOCIB Data Center, 2018). As mentioned in Sec. 2.4, netCDF files with a correction applied on the salinity and conductivity are also provided (L1_corr).

## 3.2 Gliders

To collect measurements addressing the submesoscale, two gliders were deployed on May 25 inside the study area. The coastal glider carried out measurements up to 200 m depth and the deep glider up to 500 m. The horizontal resolution was about 0.5 km for the shallow and 1 km for the deep glider. The initial sampling strategy consisted in two 50-km long, meridional tracks, 10 kilometers away one from the other, and to repeat these tracks up to 4 times during the experiment. However, due to the strong zonal currents in the frontal zone, different tracks (Fig. 6) crossing the front several times were made instead.

On May 25 at 19:24 (UTC), the deep glider payload suffered an issue with the data logging software, resulting in no data acquisition during a few hours, during which the problem was being fixed. After this event, the data acquisition could be resumed on May 26 at 08:50 (UTC).

The mean vertical separation between 2 consecutive measurements is around 16 cm. Figure 7 displays the temperature and salinity sections obtained with the 2 vehicles. The high density of measurements makes it possible to distinguish small-scale features on both sides of the front, such as strong lateral gradients, subduction or filament structures.

The gliders follow a 3-dimensional trajectory in the water column but for some specific usages it is sometimes more convenient to have the glider data as if they were a series vertical profiles. To do so, a binning is applied on the original data, leading to the L2 data, as described in Sec. 2.4.

### 3.2.1 Configuration

The information concerning the two gliders is summarised in Tab. 3. Due to safety concerns, both the deep and coastal gliders had their surfacing limited: the deep glider came to the surface one in every 3 profiles, while the coastal gliders came out one in every 10 profiles. While this strategy does not appear optimal in a scientific point of view (loss of measurements near the surface, meaning of the depth-average currents), the priority was set on the glider integrity.

**Table 3.** Characteristics of the gliders.

|  | Coastal glider | Deep glider |
|---|---|---|
| Manufacturer | Teledyne Webb Research Corp. | Teledyne Webb Research Corp. |
| Model | Slocum, G1, shallow version (200 m) | Slocum G1 Deep |
| Battery technology | Alkaline C-cell | Alkaline C-cell |
| Software version | 7.13 (navigation), 3.17 (science) | 7.13 (navigation), 3.17 (science) |
| On-board sensors | CTD (S.B.E.) | CTD (S.B.E.) |
|  | Oxygen: OPTODE 3835 (Aandera) | Oxygen: OPTODE 3830 (Aandera) |
|  | Fluorescence-Turbidity: FLNTUSLO (WetLabs) | Fluorescence-Turbidity: FLNTUSLK (WetLabs) |
| Number of casts | 160 | 392 |
| Total distance (km) | 127 | 118 |
| Max. depth (m) | 200 | 500 |

### 3.2.2 Quality control

Before the deployment, glider compass was calibrated following Merckelbach et al. (2008). The thermal-lag happening on the un-pumped Sea-Bird CTD sensors installed on the deep and coastal gliders is corrected using the procedure described in (Garau et al., 2011).

5     The checks not yet applied but planned for the next release of the Glider toolbox include: the removal of $NaN$ values, the detection of impossible dates or locations, valid ranges (depending on depth) for the variables, spikes, gradients and constant value over a large range of depths in the profiles. The tests performed that the constant value check proved useful for conductivity (and hence density and salinity). A new version of the present dataset will be released once these new checks are made 10  operational.

    Finally, oxygen concentration measurements (not shown here) seem to exhibit a lag. According to Bittig et al. (2014), this issue is also related to the time response of oxygen optodes. As far as we know, there is not yet an agreement from the community on how to correct this lag, this is why the data are kept as they are in the present version, though we don't discard an improvement of the glider toolbox to address this specific issue.

15 ### 3.3 Surface drifters

On May 25, 25 Surface Velocity Program (SVP, Lumpkin and Pazos, 2007) drifters were deployed in the frontal area in a tight square pattern with a mean distance between neighbor drifters around 3 km. In the Mediterranean Sea, they have been shown to provide information on the surface dynamics, ranging from basin scales to mesoscale features or coastal currents (Poulain et al., 2013). Almost all the drifters were equipped with a thermistor on the lower part of the buoy to measure sea 20  water temperature.

    11 out the 25 drifters, especially those deployed more to the south, were captured by the intense Algerian Current and followed a trajectory along the coast until a longitude about 5°30'E. The other drifters were deflected northward about 0°30'E, then veered northwestward or eastward and described cyclonic and anticyclonic trajectories, respectively. All the drifters moved along the front position (deduced from the SST images), until they encounter the Algerian Current (Fig. 8).

25     On average the temporal sampling resolution is close to one hour, except for 2 drifters for which the intervals are 4 and 5 hours. The velocities are directly computed from the successive positions and highlight the strength of the Algerian Current with velocities on the order of 1 m/s (Fig. 9).

### 3.3.1 Configuration

The drifters deployed during the experiment are the mini-World Ocean Circulation Experiment SVP drifters. These drifters are 30  made up of a surface buoy that includes a transmitter to relay data and a thermistor to measure the water temperature near the surface; the buoy is tethered to a holey-sock drogue centered at 15 m depth. The possible loss of the drogue is controlled with a tension sensor located below the surface buoy.

15 drifters were manufactured by Pacific Gyre and 10 by Data Buoy instrumentation (DBi). All the drifters contributed to the Mediterranean Surface Velocity Programme (MedSVP).

### 3.3.2 Quality control

Tests are applied on the position (i.e. on land), velocity and temperature records (valid ranges and spikes). Checking the platform speed is particularly relevant, as abnormally high values are intermittently encountered. See SOCIB Data Center (2018) for the threshold values used in the checks. In addition, the method developed by Rio (2012) is used to improve the accuracy of the drogue presence from wind slippage Menna et al. (2018).

## 3.4 Profiling floats

Three profiling floats were deployed in the same zone as the drifters, on May 25 (see Tab. 4). Their configuration depends on the float type: the Arvor-C has higher temporal resolution (hours) and does not go much deeper than 400 m. The A3 and Provor-bio platforms are usually set to have cycle length between 1 and 5 days, with the bio reaching maximal depth on the order of 1000 m. The floats constitute an essential tool in order to monitor the mesoscale (Sánchez-Román et al., 2017). The trajectories (Fig. 10) clearly show that profiles were acquired in the frontal area, before the floats were eventually captured by
the Algerian Current.

The Arvor-C trajectory closely follows the front position until a latitude of 36°30'N, accounting for 455 profiles in the vicinity of the front. This is probably due to its configuration: its high frequency temporal sampling makes it possible to spend more time in the near-surface layer and hence the float follows the front better than the 2 other float types. Its last profile was taken on June 14, 2014, at an approximative location of 36°15'N, $4°E$, then it drifted at the surface.

### 3.4.1 Configuration

The 3 floats provided temperature and salinity profiles thanks to the Sea-Bird CTD. In addition to these variables, the PROVBIO (PROVOR CTS4) platform measured biochemical and optical properties: colored dissolved organic matter (CDOM), chlorophyll-a concentration, backscattering (650 nm), dissolved oxygen concentration and downwelling irradiance (380, 410, 490 nm) and photosynthetically active radiation (PAR). Table 4 reports the main deployment characteristics. All the floats are manufactured
by NKE (Hennebont, France). The profiles were performed around local noon time and were used in combination with the glider measurements to study the deep chlorophyll maximum (DCM) across the front (Olita et al., 2017).

## 3.5 Current profiler

The Vessel Mounted-Acoustic Doppler Current Meter Profiler (VM-ADCP) acquired velocity profiles approximatively every 2 minutes during nighttime (22:00–6:00 UTC) at a speed of 10 knots and during the CTD surveys (see Fig. 3). The measurement accuracy is on the order of 0.01 m/s. The measurements were vertically averaged over 8 m depth bins.

**Table 4.** Characteristics of the profiling floats.

| Platform | Final date | Maximal depth (m) | Cycle length | No. of profiles | |
|---|---|---|---|---|---|
| | | | | Mission | Total |
| ARVOR-A3 | 2014-06-17 | 2000 | 1 day | 3 | 12 |
| ARVOR-C | 2014-06-17 | 400 | 1.5 hour | 144 | 2507 |
| PROVOR CTS4 | 2015-04-24 | 1000 | 1 day until June 7, then 5 days | 9 | 71 |

The velocities exhibit a dominant eastward current with speed locally larger than 1 m/s and that signal is clearly visible in the first 100 m of the water column. The velocity field is illustrated in Fig. 11 where each velocity vector is shown as a bar with a color depending on the intensity. The vertical structure is also displayed along with the front position.

### 3.5.1 Configuration

The current profiler is an Ocean Surveyor ADCP, manufactured by Teledyne RD Instruments and operating at a frequency of 150 KH. This instrument was configured with 8-m depth bins and a total of 50 bins. Final velocity profiles were averaged in 10-minute intervals. The transducer depth is approximatively 2 m.

The position and behavior (heading, pitch and roll) of the research vessel is obtained with an Ashtec 3D GPS 800 ADU positioning system that provides provide geographical positions with a 10-20 cm accuracy and heading, pitch and roll with an accuracy on the order of 1°. The instrument was calibrated to correct the misalignment angle and scaling factor. The technical report referring to this platform is available in the Annex II of Ruiz et al. (2015).

### 3.5.2 Quality checks

The vessel's velocity is one or two order or magnitudes greater than the currents that have to be measured, hence this type of current measurements requires a careful processing in order to get meaningful velocities from the raw signal. The QC procedure for the VM-ADCP is complex as it involves tests on more than 40 technical and geophysical variables (SOCIB Data Center, 2018). The different tests are based on the technical reports of Cowley et al. (2009) and Bender and DiMarco (2009), which aim primarily at ADCP mounted on moorings. The procedure can be summarised as follows:

1. Technical variables: valid ranges are checked for each of these variables: if the measurement is outside the range, the QF is set to 4 (bad data). Example of technical variables are: bottom track depth, sea water noise amplitude, correlation magnitude.

2. Vessel behaviour: its pitch, roll and and orientation angles are checked and QF are assigned based on specific ranges. In addition the vessel velocity is checked and anomalously high values are also flagged as bad.

3. Velocities: valid ranges are provided for the computed current velocities: up to 2 m/s, velocities considered as good; between 2 and 3 m/s, probably good, and above 3 m/s, bad.

The application of all these tests lead to Fig 12, which illustrates the QF during the whole mission. The 3 main periods during which the ADCP was turned off are shown as grey areas. In addition, no measurements are available in the first meters of the water column, due to the position of the ADCP on the ship, at a depth of approximately 2 m.

Overall the quality of the data tends to deteriorate when the depth increases, as reflected by the bad and missing values. In the first 200 m, about 95% of the measurements are considered as good. Below 200 m, the ratio drops to 57% with more than 21% of missing values. Note that the flags 5 (, 7 and 8 were not used in this case but kept in the plot.

### 3.6 Nutrients

Samples for nutrient analysis were collected in triplicate from CTD Niskin bottles and immediately frozen for subsequent analysis at the laboratory. Concentrations of dissolved nutrients (Nitrite: $NO_2^-$, Nitrate: $NO_3^-$ and Phosphate: $PO_4^{3-}$ were determined with an autoanalyzer (Alliance Futura) using colorimetric techniques (Grasshoff et al., 1983). The accuracy of the analysis was established using Coastal Seawater Reference Material for Nutrients (MOOS-1, NRCCNRC), resulting in recoveries of 97%, 95% and 100% for $NO_2^-$, $NO_3^-$ and $PO_4^{3-}$, respectively. Detection limits were $NO_2^-$:0.005 $\mu$M, $NO_3^-$: 0.1 $\mu$M and $PO_4^{3-}$: 0.1 $\mu$M.

## 4 Description of the database

The AlborEx mission generated a large amount of data in a region sparsely sampled in the past. The synergy between lower-resolution (CTD, drifters, floats) and high-resolution data (ADCP, gliders) makes this dataset unique for the study of submesoscale processes in the Mediterranean Sea. Moreover its multidisciplinary nature makes it suitable to study the interactions between the physical conditions and the biogeochemical variables.

### 4.1 File format and organisation

The original data files (i.e. obtained directly from the sensors and with a format depending on the manufacturer) are converted to Network Common Data Form (netCDF, https://doi.org/http://doi.org/10.5065/D6H70CW6, last accessed on August 3, 2018), an Open Geospatial Consortium (OGC) standard widely adopted in atmospheric and oceanic sciences. Each file contains the measurements acquired by the sensors as well the metadata (mission name, principal investigator, . . . ). The structure of the files follows the Climate and Forecast (CF) conventions (Domenico and Nativi, 2013) and are based on the model of OceanSITES (Send et al., 2010).

### 4.2 File naming

In order to keep the file names consistent with the original database, it is decided to keep the same file names as those assigned by SOCIB Data Center. Let us decompose one file name into its different parts:

```
dep0007_socib-rv-scb-sbe9002_L1_2014-05-25.nc
```

`dep0007` indicates the number of the deployment, where deployment is the equivalent to the start of a mission or survey with a given platform. The deployment ends when the mission is over or if the platform stops acquiring data.

`socib-rv` is the code for the platform, in this case the SOCIB coastal research vessel.

5 `scb-sbe9002` is the instrument identifier, here the CTD SeaBird 9Plus. Note that the instrument is described in the metadata of the netCDF file.

`L1` is the processing level (see Sec. 2.4).

`2014-05-25` is the deployment date (year-month-day).

Now the general naming is defined, Tab. 5 list below the different files made available in the dataset.

**Table 5.** Platform corresponding to the different files.

| File name | Platform |
|---|---|
| `dep0023_socib-rv_scb-rdi001_L1_2014-05.nc` | ADCP |
| `dep0007_socib-rv_scb-sbe9002_L1_2014-05-25.nc` | CTD |
| `dep0001_drifter-svp***_scb-svp***_L1_2014-05-25.nc` | SVP drifers ($\times$ 25) |
| `dep0005_icoast00_ime-slcost000_L1_2014-05-25_data_dt.nc` | Coastal glider |
| `dep0012_ideep00_ime-sldeep000_L1_2014-05-25_data_dt.nc` | Deep glider |
| `dep0001_profiler-drifter-arvora3001_ogs-arvora3001_L1_2014-05-25.nc` | Arvor-A3 float |
| `dep0001_profiler-drifter-arvorc_socib_arvorc_L1_2014-05-25.nc` | Arvor-C float |
| `dep0001_profiler-drifter-provbiol1001_ogs-provbiol1001_L1_2014-05-25.nc` | Provor-Bio float |
| `dep0015_socib-rv_scb-met009_L1_2014-05-25.nc` | Weather onboard R/V |
| `dep0015_socib-rv_scb-pos001_L1_2014-05-25.nc` | Navigation data from R/V |
| `dep0015_socib-rv_scb-tsl001_L1_2014-05-25.nc` | Thermosalinograph |
| `dep0015_socib-rv_scb-tsl001_L1_2014-05-25_HR.nc` | Thermosalinograph (high-res.) |

`***` in the file names stands for 3 digits.

## 10 4.3 Data reading and visualisation

The standard format (netCDF) in which the data files are written makes the reading and visualisation straightforward. A variety of software tools such as ncview, ncBrowse or Panoply are designed to visualised gridded fields. Here the data provided consist of trajectories (surface or 3D), profiles, trajectory-profile, which can be easily read using the netCDF library in different languages (Tab. 6).

15 Examples of reading and plotting functions, written in Python, are also provided (Troupin, 2018). They allow users or readers to get the data from the files and reproduce the same figures as in the paper, constituting a good starting point to carry out further specific analysis.

**Table 6.** NetCDF libraries for various languages.

| Programming language | Library |
| --- | --- |
| Python | https://github.com/Unidata/netcdf4-python |
| Fortran | https://github.com/Unidata/netcdf-fortran |
| C | https://github.com/Unidata/netcdf-c |
| Javascript | https://www.npmjs.com/package/netcdf4 |
| Octave | https://github.com/Alexander-Barth/octave-netcdf |
| Julia | https://github.com/Alexander-Barth/NCDatasets.jl |
| MATLAB | Native support since version R2010b |

When accessing the data catalog, users are provided a list of in-house visualisation tools designed to offer quick visualisation of the file content. The visualisation tools depend on the type of data: *JWebChart* is used for time series; *Dapp* displays the trajectory of a moving platform on a map; the *profile-viewer* allows the user to select locations on the map and view the corresponding profiles.

## 5 Conclusions and perspectives

The AlborEx observations acquired in May 2014 constitutes a unique observational data set that captured mesoscale and submesocale features in a particularly energetic frontal zone in the western Mediterranean Sea. The potential uses of the dataset can be separated in different topics:

– Hydrodynamics model validation: with their increasing resolution, models are becoming able to properly reproduce small-scale structures, but the correct timing and location of these features remain a challenging topic.

– High-resolution remote-sensing data validation: high quality in situ measurements of the sea surface are essential for the validation of operational product such SST or Ocean Color.

– Study of mechanisms: the Mediterranean Sea is often referred to as a laboratory for oceanography and in particular the Alboran Sea is the stage of intense processes of mixing, subduction and instabilities.

– Assessment of mechanisms responsible for intense vertical motions.

The version of the dataset described in the present paper contains files that have been processed and standardised so that they are directly usable by scientists without having to perform unit or format conversions from the manufacturer raw data files.

Updates will be performed when new versions of the files or new files are made available.

## 6 Code and data availability

Following SOCIB general policy, the data are made available as netCDF files through the SOCIB Thematic Real-time Environmental Distributed Data Services (THREDDS) Data Server, a standard way to distribute metadata and data using a variety of remote data access protocols such as OPeNDAP (https://www.opendap.org), Web Map Service (WMS) or direct HTTP access. In addition, the whole AlborEx dataset has been assigned a Digital Object Identifier (DOI) to make them it and uniquely citable. The most recent version of the dataset is accessible from http://doi.org/10.25704/z5y2-qpye and the nutrient data, in process of being included in the catalog, are available at https://repository.socib.es:8643/repository/entry/show?entryid= 07ebf505-bd27-4ae5-aa43-c4d1c85dd500.

Upgrades will be performed periodically with the implementation of fresh or better QCs on sensors such as the ADCP, CTD or gliders. The new releases will be available using the same Zenodo identifier, but with be assigned a different version number, each version having its own DOI. Files not available at the time of the writing will also be appended to the original database.

Concerning the improvement of the quality control, it is worth mentioning the new tests that will be implemented in the SOCIB Glider Toolbox (Troupin et al., 2016).

The checks performed on the ADCP velocities involve a set of parameters that can also be fine-tuned to improve the relevance of the quality flags. Nevertheless, noticeable changes are not expected with respect to the quality flags displayed in Fig. 12.

Finally, the quality of the CTD and the glider profiles can be improved by using the salinity measurements of water samples collected during the mission. This type of correction might not be essential for the study of mesoscale processes but is crucial when one is focused on long-term studies and when a drift can be observed in the salinity measurements.

A set of programs in Python to read the files and represent their content as in the figures presented through the paper are available at https://github.com/ctroupin/AlborEX-Data. The programs are written in the form of documented Jupyter notebooks, a web application that combines code fragment, equations, graphics and explanatory text (http://jupyter.org/, last accessed 14 August, 2018) so that they can be run step by step. The figures colormaps were produced using the `cmocean` module (Thyng et al., 2016).

*Author contributions.* C.T. prepared the figures and the first version of the manuscript. A.P. and S.R. edited the manuscript. J.G.F. and M.A.R. lead the data management and the creation of a DOI in Datacite. C.M. implemented the processing of the ADCP data. G.N. helped with the processing of the drifters and profiling floats. E.A. and A.T. processed and provided the biochemical data. I.R. finalised the QC documentation.

*Competing interests.* The authors declare that the research was conducted in the absence of any commercial or financial relationships that could be construed as a potential conflict of interest.

*Disclaimer.* The authors do not accept any liability for the correctness and appropriate interpretation of the data or their suitability for any use.

5    *Acknowledgements.* AlborEx was conducted in the framework of PERSEUS EU-funded project (Grant agreement no. 287600). Glider operations were partially funded by JERICO FP7 project. AP acknowledges support from the Spanish National Research Program (E-MOTION/CTM2012-31014 and PRE-SWOT/CTM2016-78607-P). SR and AP are also supported by the Copernicus Marine Environment Monitoring Service (CMEMS) MedSUB project. AO was supported by the Jerico-TNA program, through the FRIPP (FRontal Dynamics Influencing Primary Production) project. The profiling floats and some drifters were contributed by the Argo-Italy program. The proceedings of such an ambitious mission would not have been possible without the involvement of a numerous staff both at sea and on land: A. Massanet, M. Palmer, I. Lizaran, C. Castilla, P. Balaguer, M. Menna, K. Sebastián, S. Lora, and A. Bussani .

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

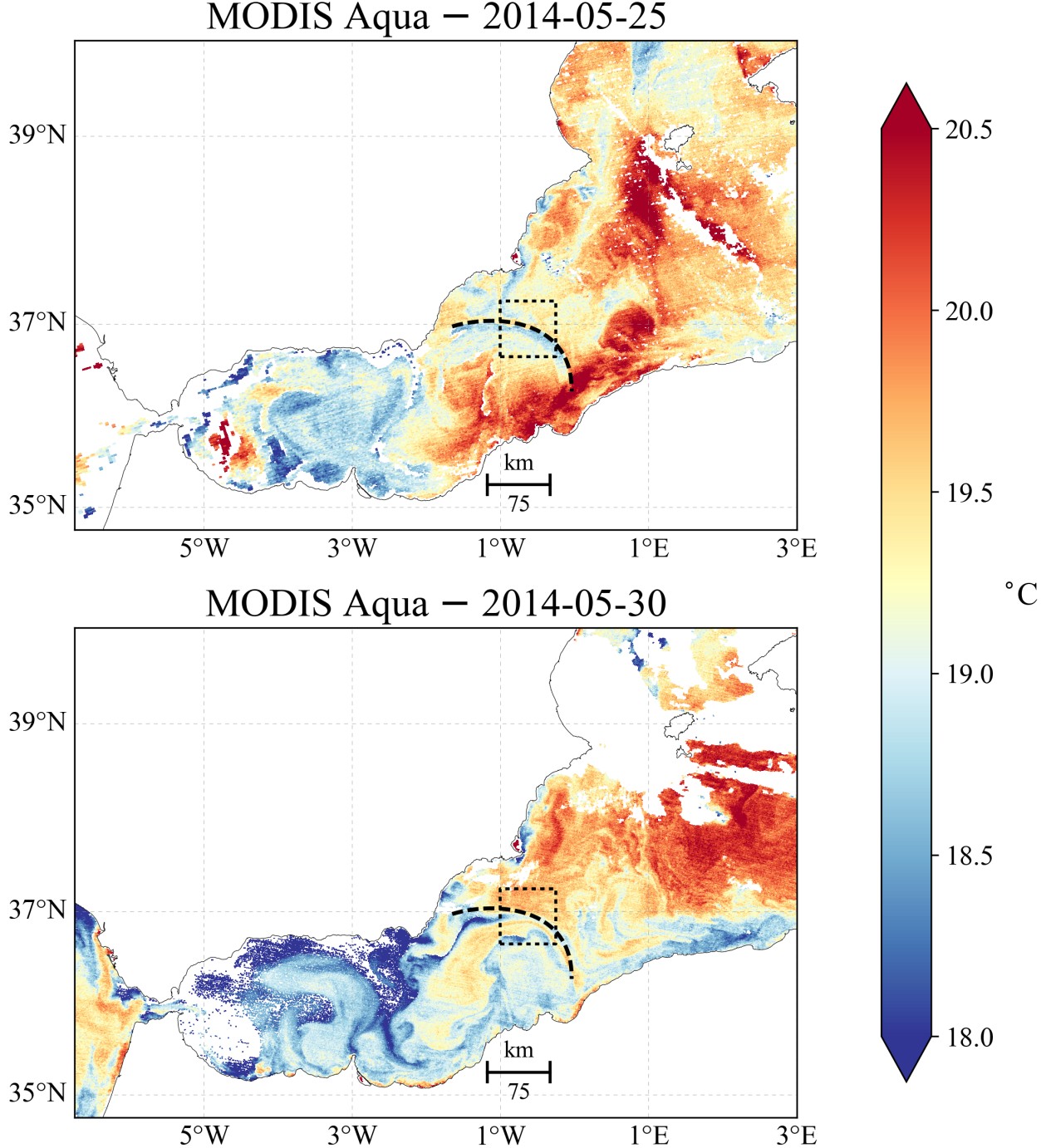

**Figure 2.** Sea surface temperature in the western Mediterranean Sea from MODIS sensor onboard Aqua satellite corresponding to May 25 and 30, 2014. The dashed black line indicates the approximative position of the front based on the temperature gradient for the period 25–30 May. Level-2, 11 $\mu m$, night-time images were selected. Only pixels with a quality flag equal to 1 (good data) were conserved and represented on the map. The same front position is used in the subsequent figures.

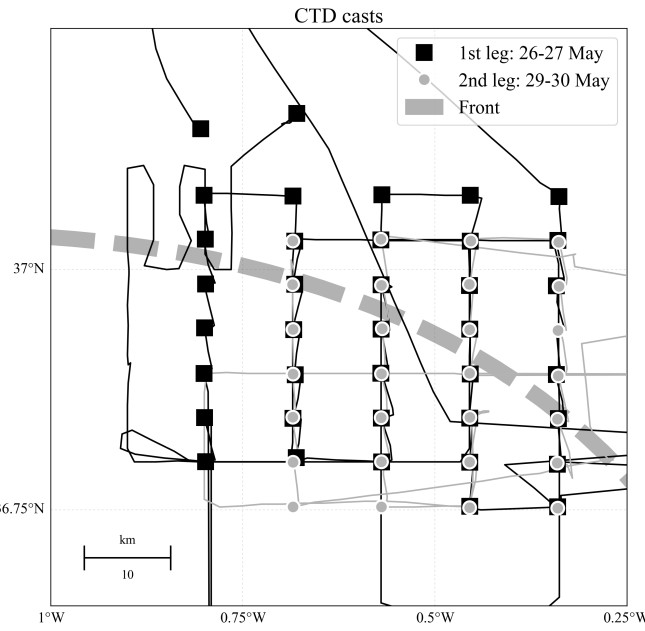

**Figure 3.** The CTD casts were organised in 5 legs that crossed the front and were repeated over 2 periods, at the beginning and the end of the mission..

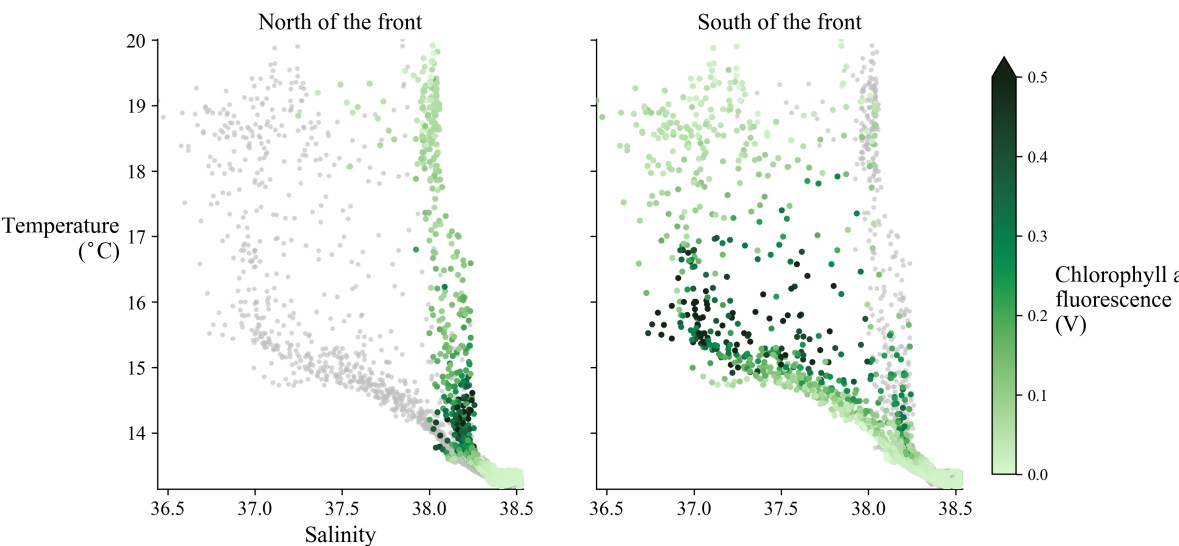

**Figure 4.** The T-S diagrams are shown separately for the casts located north and south of the front (broad, dashed line) .

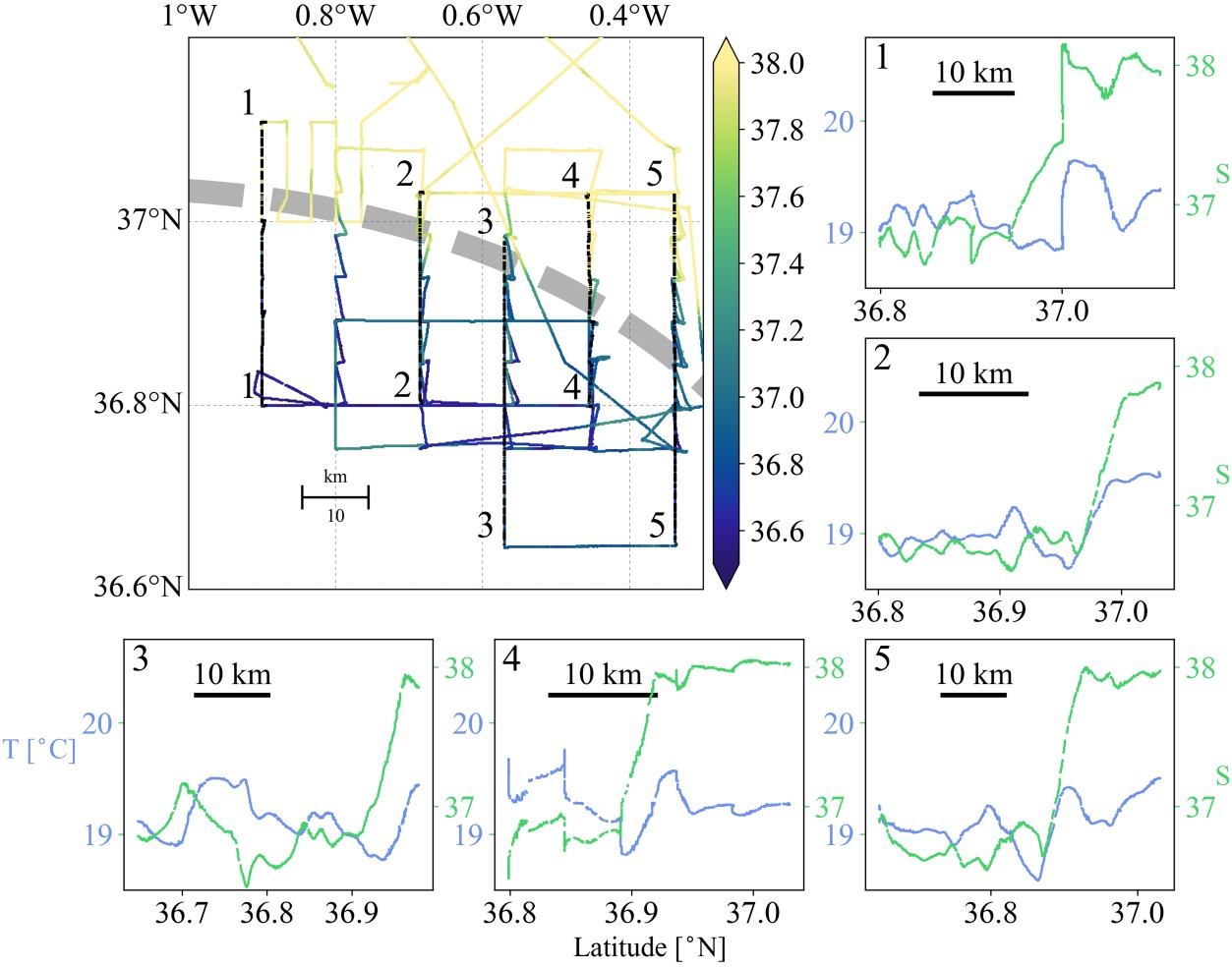

**Figure 5.** The near-surface salinity (colored dots) measured by the thermosalinograph evidences the strong horizontal gradients, in agreement with the front position as obtained using the SST (broad, dashed line). The 5 subplots depict the temperature and salinity along select meridional tracks.

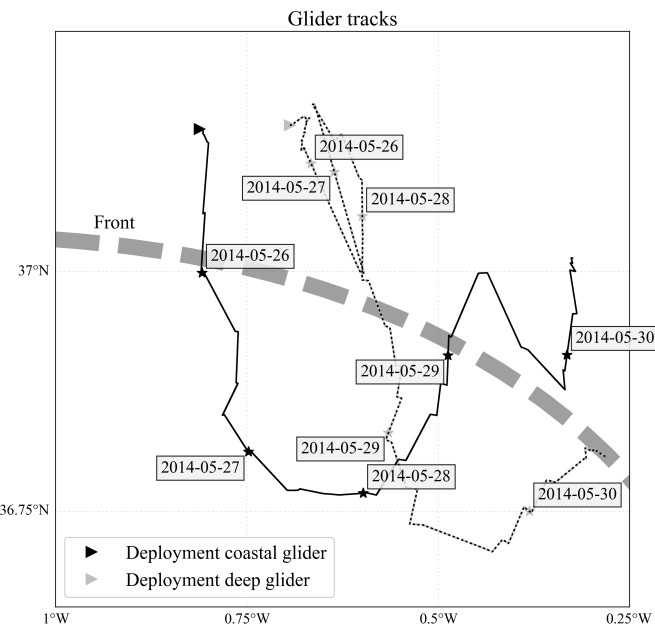

**Figure 6.** Deployment positions and trajectories of the gliders. Different time instances separated by one day are indicated on the tracks to provide a temporal dimension.

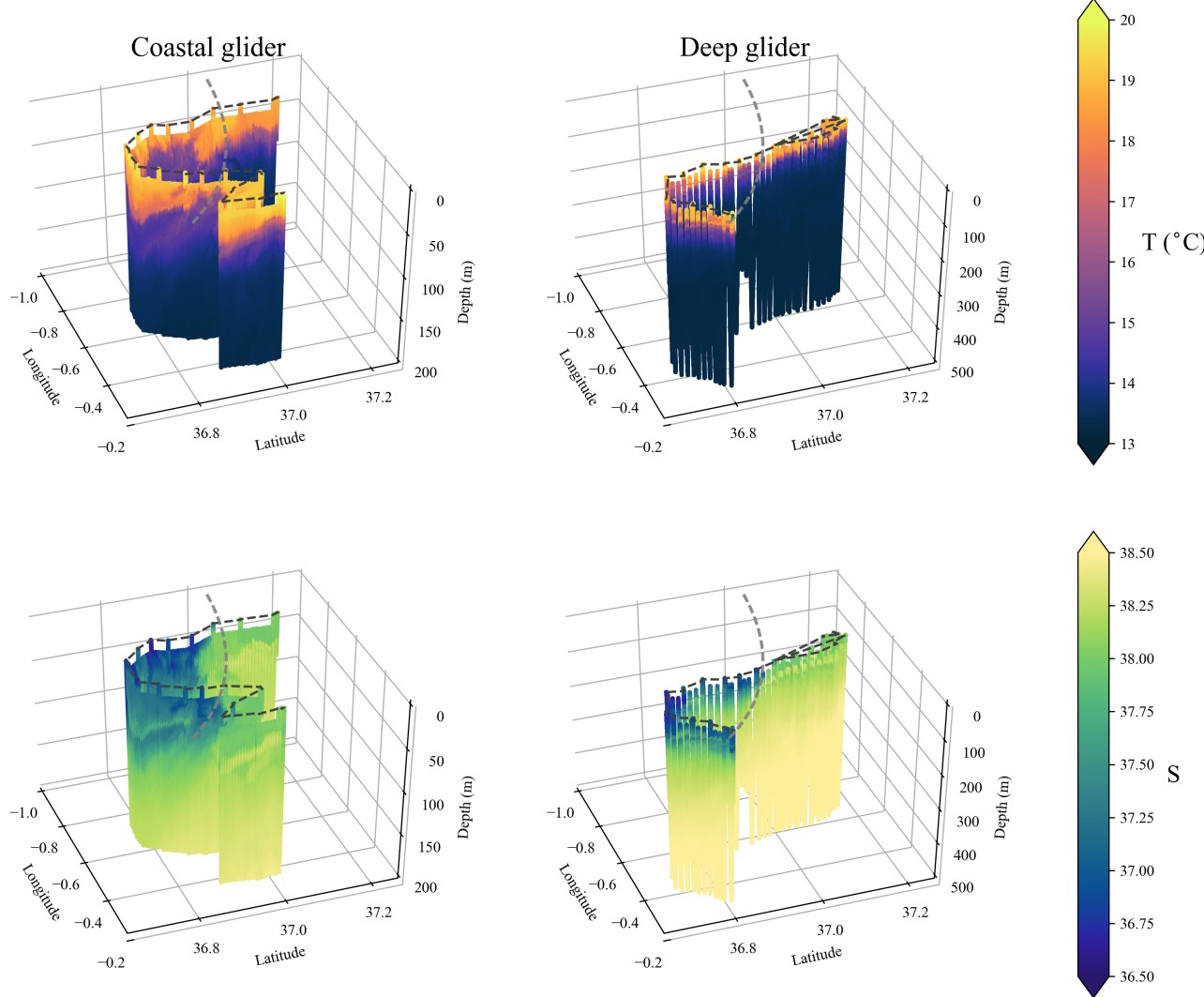

**Figure 7.** Temperature (top) and salinity measured by the two gliders. The approximative front position at the surface is shown as a dashed, grey line.

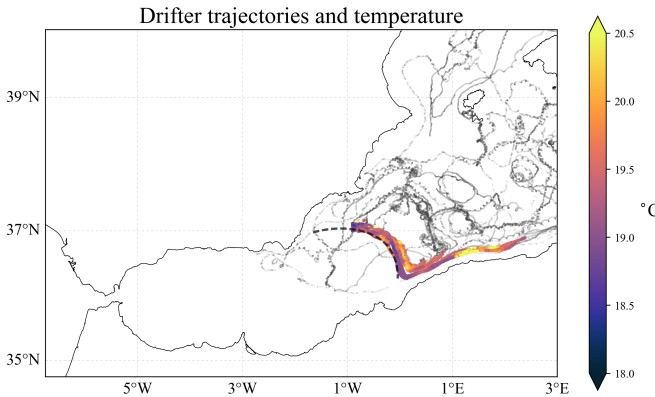

**Figure 8.** Surface drifter trajectories. For the sake of simplicity and clarity, the temperature, when available, is only shown for the duration of the AlborEx mission (May 25-31, 2014).

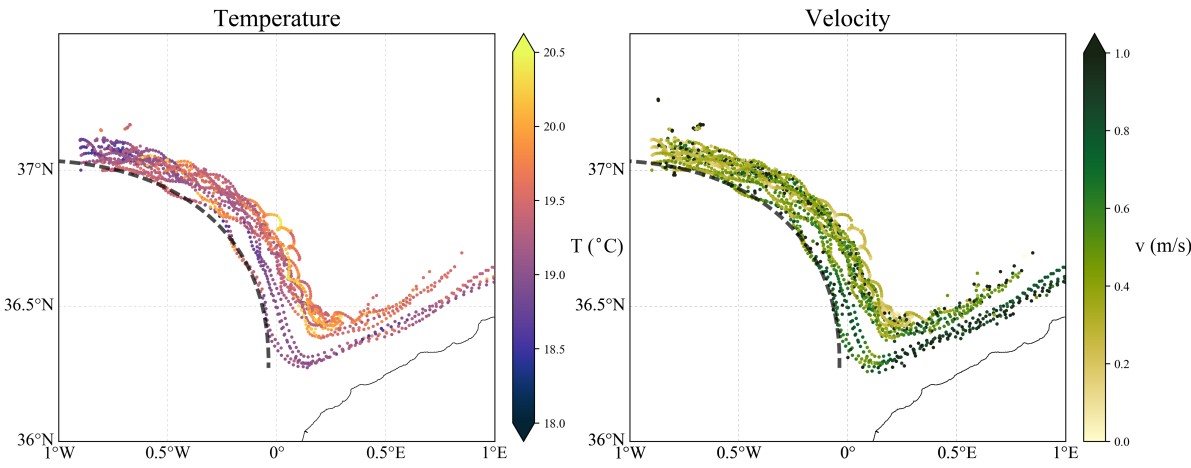

**Figure 9.** Drifter temperature (left-hand side) and velocity in the area of study.

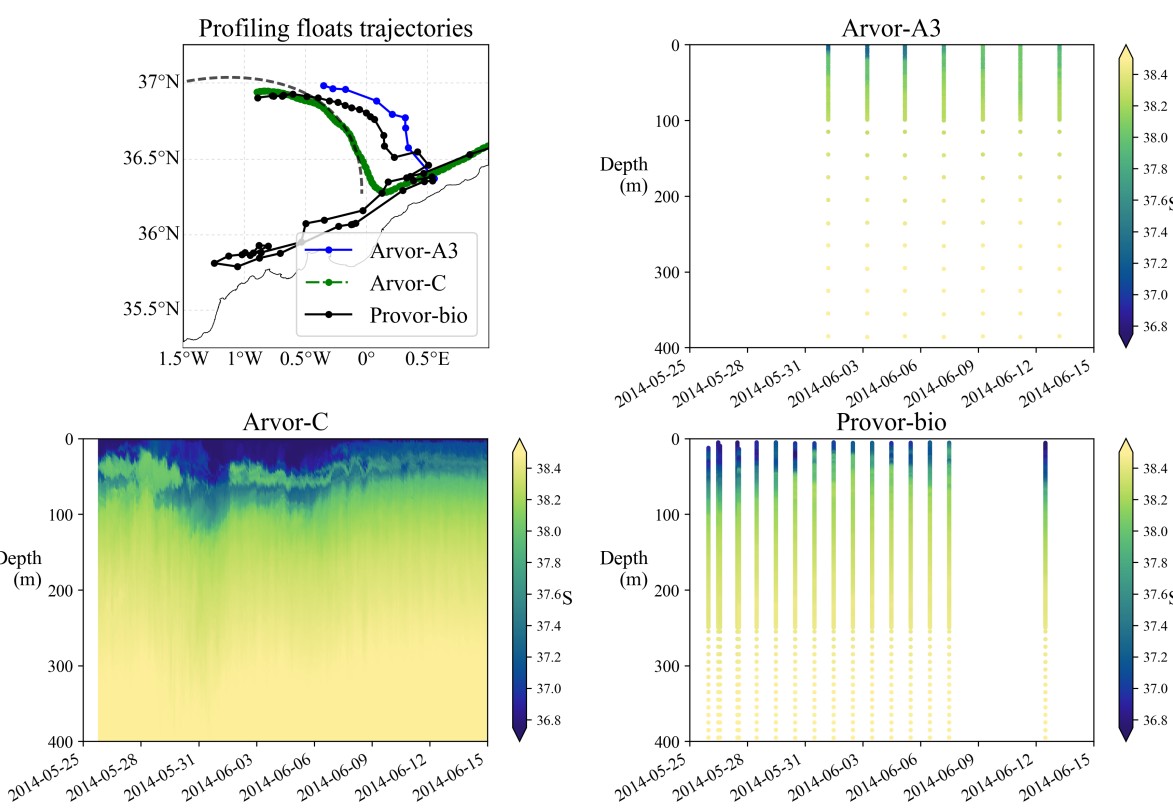

**Figure 10.** Profiling floats trajectories (top-left panel) and salinity from May 25 to June 15, 2014.

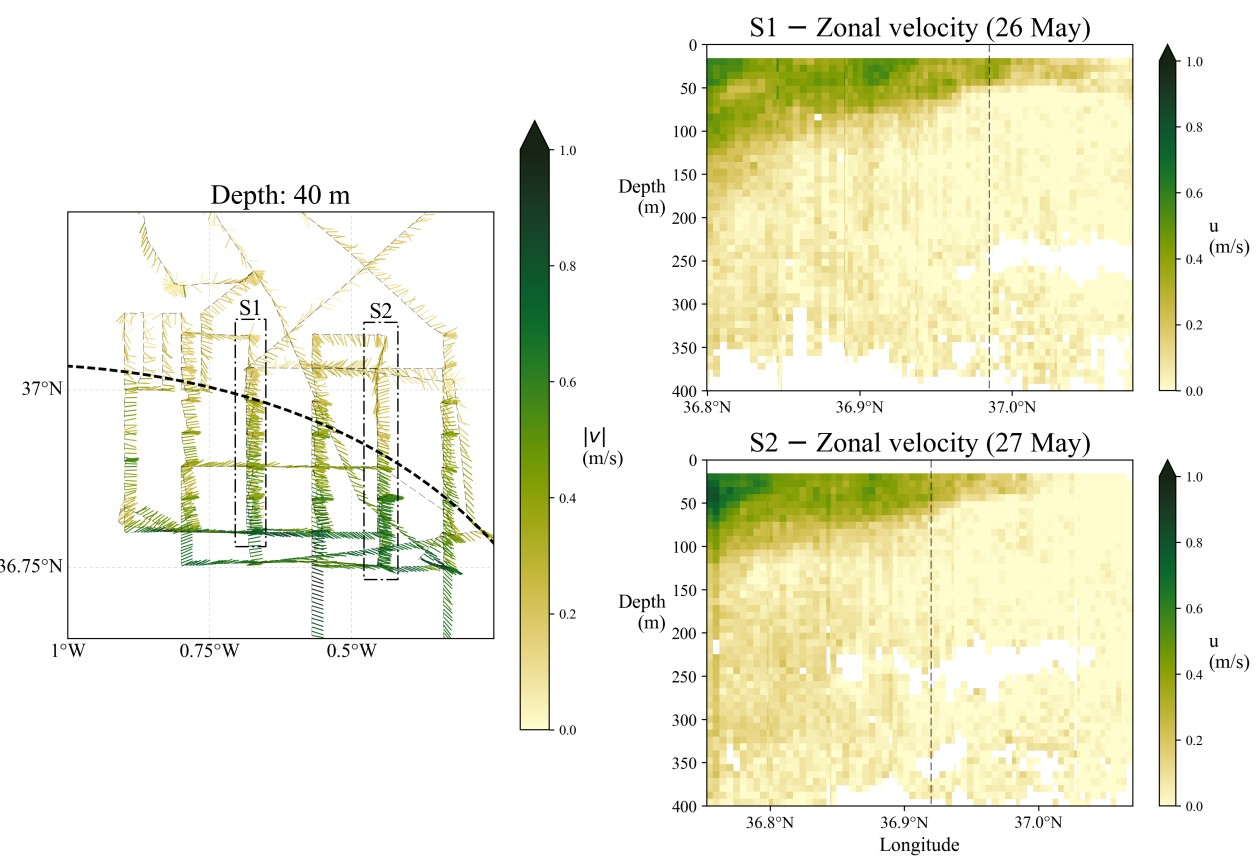

**Figure 11.** Velocity field obtained with the ADCP at a 40 m depth (left panel) and sections of zonal velocity on May 26 (S1) and 27 (S2). The locations of the sections are indicated by dashed rectangles on the map. Only data with a quality flag equal to 1 (good data) are represented

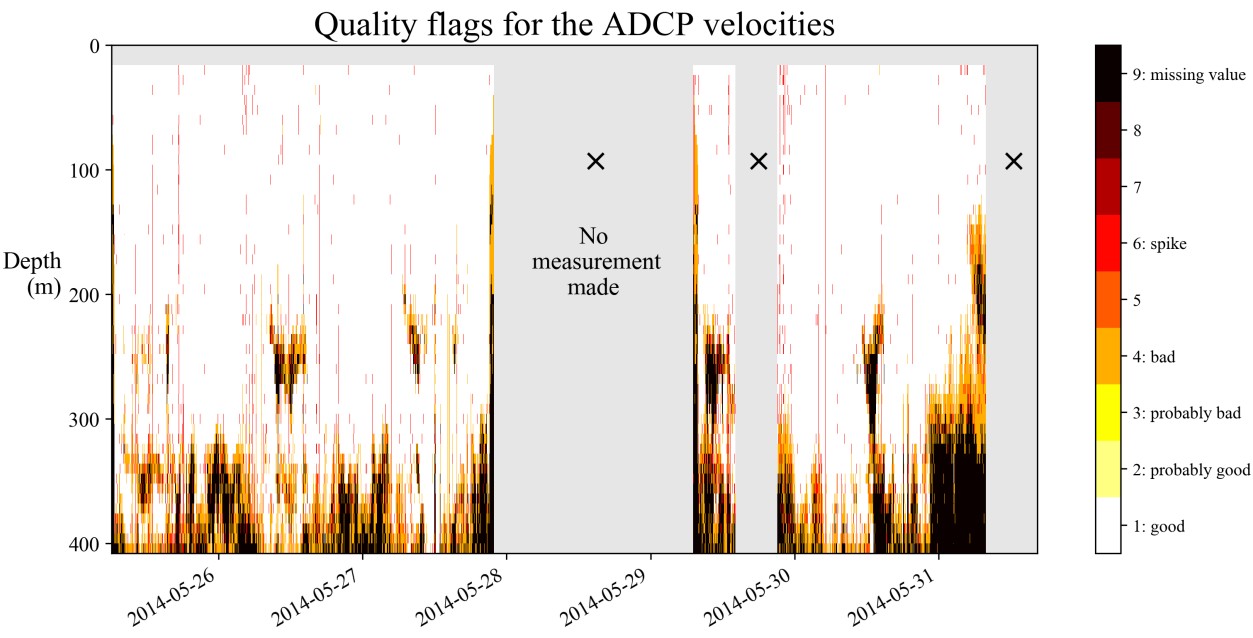

**Figure 12.** Quality flags for the velocity measurements. The areas marked with a × are those during which the VM-ADCP was no acquiring measurements.