# Peer review of "The AlborEX dataset: sampling of submesoscale features in the Alboran Sea"

_Earth System Science Data, 2018_

## Referee Comment (RC1) · Anonymous Referee #1 · 13 Sep 2018

This article (categorized as "review") by Troupin et al. is addressing a multidisciplinary data set collected in the western Mediterranean Sea during the AlborEX campaign. During the campaign in-situ observing devices (ships, floats, gliders, drifters. . .) have been used (described here) but also satellite data. In the manuscript some aspects of the data set are described. As it stands now I do not recommend publication in ESSD.

For the review I followed the ESSD evaluation criteria and also considered the general scope of the journal (as described on the website).
First - Is this a "review" article? ESSD defines review articles as:
". . .may compare methods or relative merits of data sets, the fitness of individual methods or data sets for specific purposes, or how combinations might be used as

more complex methods or reference data collections."

As I read it from the manuscript this is not the case. The current version of the manuscript reads more as a copy of data information from individual reports and the data section in scientific publications related to the experiment. As it stands, I do not see the criteria for a "review" type article fulfilled.

**Significance**

Three sub-criteria to evaluate:

- Uniqueness: It should not be possible to replicate the experiment or observation on a routine basis. Thus, any data set on a variable supposed or suspected to reflect changes in the Earth system deserves to be considered unique. This is also the case for cost-intensive data sets which will not be replicated due to financial reasons. A new or improved method should not be trivial or obvious.
  **The data set is unique.**
  (rating: 1 Excellent)

- Usefulness: It should be plausible that the data, alone or in combination with other data sets, can be used in future interpretations, for the comparison to model output or to verify other experiments or observations. Other possible uses mentioned by the authors will be considered.
  **The current manuscript does not provide information that promote the reuse of the data set (it may for subsets). No attempt is made to provide a structured overview about the workflow that is linke dot the creation of the data set and, equally important, the QA/QC are not provided in a transparent way. For example, in the netcdf data files I see different QC flags provided – one is for example "SOCIB Quality control Data Protocol". What does that mean? This is not an international standard. A data set descrip-**

[Figure]

tion, as envisioned in this ESSD submission, should exactly describe such
non-standard QC procedures. Which QA and QC methods were applied
(give brief description, DOIs if applicable)?
I also miss any information how/if this data is disseminated via interna-
tional data centres and how the data QC and dissemination is coordinate
with the respective observing networks (Argo, DBCP, . . .). Seadatanet is
been mentioned in the text but it is unclear which specific recommenda-
tions are given.
(rating: 4 poor)

- Completeness: A data set or collection must not be split intentionally, for exam-
  ple, to increase the possible number of publications. It should contain all data that
  can be reviewed without unnecessary increase of workload and can be reused in
  another context by a reader.
  **It is difficult to evaluate this point. However, the nutrient data is not men-
  tioned but is, according to Pascual et al. 2017 part of the AlborEX cam-
  paign. I would expect that these data set are described here as well (and
  respective QC (e.g. GO-SHIP nutrient manual??) and associated uncer-
  tainty estimates**
  (rating: 2 to 3)

**Data quality**
The data must be presented readily and accessible for inspection and analysis to
make the reviewer's task possible. Even if a data set submitted is the first ever
published (on a parameter, in a region, etc.), its claimed accuracy, the instrumentation
employed, and methods of processing should reflect the "state of the art" or "best
practices". Considering all conditions and influences presented in the article, these
claims and factors must be mutually consistent. The reviewer will then apply his or her
expert knowledge and operational experience in the specific field to perform tests (e.g.
statistical tests) and cast judgement on whether the claimed findings and its factors –
individually and as a whole – are plausible and do not contain detectable faults.

**I touched on that already under "Usefulness". In the manuscript no transparent QC assessment is presented. What were the methods of processing (provide key steps, DOI at least). What were, including quantification of uncertainties and qualification via flags, the results of the QA/QC procedures? Which were the major shortcomings of the data acquisition process and what could be done better in the future? For example, has the drifter data included in the European E–SurfMar data base and also in the DBCP global drifter data sets? Have the recommendations (Best Practices, Protocols) from E–SurfMar / DBCP considered? It looks like no commonly agreed standard has been used for some paramters – as "SOCIB Quality control Data Protocol" suggest?**
(rating: 3)

**Presentation quality**
Long articles are not expected. Regarding the style, the aim is to develop stereotypical wording so that unambiguous meaning can be expressed and understood without much effort. The article should express clearly what has been found, where, when, and how. The article text and references should contain all information necessary to evaluate all claims about the data set or collection, whether the claims are explicitly written down in the article, or implicit, through the data being published or their metadata. The authors should point to suitable software or services for simple visualization and analysis, keeping in mind that neither the reviewer nor the casual "reader" will install or pay for it.
**mostly OK (given the limitation outlined in the previous points). It would be useful to include a brief introduction into the "design of the experiment. Visualisation tools are not given.**
(rating: 2-3)

**Specific comments:**

**P2/l.4: I do not agree with the statement: "a perfect observational system would consist in dense array of sensors present at many geographical locations, many depths and measuring almost continuously a wide range of parameters…" – this "generalization" is trivial and useless. From an observing design point of view a "perfect" observing system must follow a design that will record only the observations that are needed to analyse the problem. As such the perfect observational system always depends on motivation for the experiment (or the problem in more general words) - in some cases a "perfect observing system" may comprise only one single sensor at one single depth at different locations if this has been found a sufficient approach for solving the problem (e.g. estimating global warming through a global tomography array). Please reformulate the statement along those lines.**

---

## Referee Comment (RC2) · Anonymous Referee #2 · 24 Sep 2018

—- GENERAL COMMENTS —-

Please find below my review of the manuscript entitled "The AlborEX dataset: sampling of submesoscale features in the Alboran Sea" by Troupin et al. I think the data and the paper are relatively well presented. I especially enjoyed that all the files are netCDF format. While the data are limited to a very local application (a 6-day experiment from one sub-region of the Mediterranean Sea), the data are in high-quality and may be useful for process-related studies. Overall, the manuscript may be suitable for publication after moderate reviews. This decision is detailed below.

—- MAJOR COMMENTS —-

My major concerns on the actual version of the paper are the following:

[Figure]

1. I think the text is not well organized. Some info on the data is find in Section 2 (AlborEX mission) and in Section 3.3 (Data Processing). This spreading of information makes the search for information through the paper difficult. I would bring Section 3.3. earlier in the paper and avoid to spread the information for each platform in different sections. Some specific comments below are related to this problem (e.g. mention of flags even before introducing them).

2. The QC control is a weakness in this manuscript as it suggests that some QC is done, but it is not very clear on which data and how it is done. For some instruments, QC flags and their meaning are embedded in the files (e.g. float and drifters), but some doesn't (glider files). This inconsistency is not so much a problem to me as long as it is clearly stated in the paper which files contains QC flags. These quality flags should however be defined in the text. There are several mentions of "quality flags" in the text and figure caption, but little explanation is provided on these. Figure 12 has 9 quality flags that are not even described (although I see their meaning in drifters and float files). Where the QC is easy to reference (e.g. "file generated with Socib glider toolbox vX.X", or "File QC done using Socib standard procedure following a procedure described in a certain paper", etc.), it should be mention in the netCDF file as well.

3. Why all processing level are not provided? The text suggests that all levels are provided (e.g. Table 3), but at the moment mostly L1 is provided. For gliders, L1 and L2 are provided. For the Float, L1 is provided for Arvor-A3 and Provor-Bio, but L0 for Arvor-C. Why? No explanation for this is provided (I think float data should be provided in L1 and L2 level as well). If some QC is applied on L1, maybe L0 should be provided as well to the future user? For glider L2 data, a choice is made regarding the vertical binning of the profiles. Which size these vertical bins are? This information should be provided somewhere.

4. Nowhere the sensor configurations are specified. I think a table gathering this information is worth it. For each platform, the list of sensor should be presented with their configuration (sampling frequency, ADCP ping-per-ensemble, ADCP vertical bin

size, etc.). This should include all variables collected, for example, from the ship meteo station from which little information (or none) is present in the text. Same for the glider where there is Chl-a and turbidity data in the files, but these were not mentioned in the text. A table gathering this information would be useful.

5. A table regrouping all the platform with their basic configuration as well as their number of casts (when it applies) should be provided (sort of extended Table 3).

—- TEXT-SPECIFIC COMMENTS —-

- Figure 1 too small (should take page width)

- Figure 2 too small (should take page width)

- Figure 2 caption: there is mention of "flag data equal to 1" while these flag are not introduced in the text.

- p.7, L1: The "total number of valid measurement" is not very useful. I would rather put the number of valid casts (see comment above on a new table with this info).

- p.7, L6: "a spatial interpolation is applied on the original data, leading to the so-called Level-2 data, further described in Sec. 3.3." What does 'spatial interpolation' means? Section 3.3 is not very explicit on this. I know you mean that the glider yos have been separated into downward and upward casts and then assigned to a geographical coordinate, but maybe this should be stated explicitly (and I don't think "spatial interpolation" is an accurate description). Moreover, Is there any vertical interpolation done? Because there are still some NaNs in L2 data.

- p.7, L15: "Interestingly, all the drifters exhibit a trajectory close to the front position" -> Not clear what "trajectory close to the front means". Moreover, is that really surprising that surface drifter would aggregate on a front?

- Figure 8 caption: "for the duration of the mission" -> You mean the ship mission? Or the AlborEX campaign?

- Figure 10: plots on the right column are of little information here (too low resolution to mean something), I would remove.

- Table 1: "Period" should be replaced by "cycle length" as referred to in the text (Section 2.2.4).

- Table 1: netCDF file for Provor-bio indicates deployment end date 2015-04-24T12:02:59+00:00, which is different from this table.

- Figure 11 caption: "quality flag" not defined.

- Section 3.3.1: A Section on processing levels, but they are not all provided. Why? I think all levels should be provided. This is related to a previous comment.

- p.14, Level 2 (L2): "obtained by interpolating the L1 data" -> How L2 is obtained by "interpolating" L1? Isn't L1 cut into casts that makes L2?

- p.14, Level 2 (L2): "It is only provided for gliders, mostly for visualization and post-processing purposes: specific tools designed to read and display profiler data can then be used the same way for gliders." -> Is there a problem with this sentence? I don't understand it.

- Section 3.3.1 / Table 3: Is L1 level for float equivalent to L2 level for glider? For consistency, I think profiling float should have L1 and L2 data as well since these instruments have similarities on the way they profile the water column...

- p.12, L1: "This type of current measurements requires a careful processing in order to get meaningful velocities from the raw signal" -> Why? What are the limitations that makes this instrument more sensitive compare to other ones?

- p.12, L4: "Figure 12 shows the QF during the whole mission." -> How QF are calculated?

- Figure 12: Too small.

- Figure 12 and text below: 9 different quality flag are presented without any introduction on how they are calculated.

- Section 3.3.2 is very short. Should be re-worked following comments above.

—- COMMENTS ON DATA FILES —-

The dataset consists of a relatively large number of files. I did my best but it was nearly imposisble to review them all in details. Here are some comments:

- There are very large spikes in deep glider turbidity

- There are missing data for about 10h in deep glider data between May 25-26. Unless I missed it, no explanation for this are provided.

- Oxygen data for both glider seems to suffer from thermal lag problems

- Provor-bio datafile contains levels down to over 7000m. Some problems are found: 1. Why such long level dimension? 2. No good data is found below $\sim$325m, although Table 1 suggest that the float is profiling to 1000m

- Arvor A3 data file suffers from similar problem: file contains data only down to 115m while Table 1 says 2000m

- Arvor-C data file (only L0 provided) do not contain metadata (no file attributes, etc.). In addition, missing data (at least for temperature) appears to me as very large numbers (9.969210e+36) that makes them difficult to manipulate.

- R/V Socib CTD and thermosalinograph files say that units of temperature are "C". I prefer the convention from glider files which uses "Celsius".

—- MINOR COMMENTS —-

- p.2; L23: "makes it possible" -> makes possible - p.2; L23: "creation and publication of aggregated datasets covering the Mediterranean Sea" -> SeaDataNet is not only about the Mediterranean - p.2; L32: "thanks due to" -> thanks to - Section

2.2.1: "CTD surveys" or CTD legs? - Glider L1 files (e.g. dep0012_ideep00_ime-sldeep000_L1_2014-05-25_data_dt.nc) say that the project is "PERSEUS". Is that right? There is no mention of the AlborEX project in the file header. - p.10, L1: problems with latitude longitude degree symbol. - p.10, L5: temperature, salinity and T,S is use on the same line. Please homogenize. - p.12, L17: "Network Common Data Form (netCDF, https://doi.org/(http://doi.org/10.5065/D6H70CW6, last accessed on August 3, 2018)" Is there a mis-placed parenthesis? - p.13, L2: problem with file name (too long for page) - p.16, L25: How stable in time the python codes made available on Github will be?

---

## Referee Comment (RC3) · Anonymous Referee #3 · 2 Oct 2018

The manuscript describes a dataset collected during the AlborEX experiment in the Western Mediterranean in May 2014. The experiment aimed at studying submesoscale dynamics and its impacts on phytoplankton. Several platforms have been used: ship-borne measurements (CTD and ADCP), autonomous profiling platforms (3 floats, 2 gliders), and surface drifters.

The manuscript is illustrated by useful figures of good quality. The dataset consists in a nice combination of platforms collecting data across a submesoscale front. This is dataset led to several publications. I still have major concerns regarding its publication in the present form.

Thus, I recommend its publication Earth System Science Data after major revisions considering my following comments.

[Figure]

Major comments:

- What are the instruments specifications? A list of the parameters measured by each platform along with the corresponding sensor name must be provided for the CTD, glider and profiling floats.

- Were they any water sample taken during the cruise in order to calibrate the CTD, or chlorophyll-a fluorescence? More than four years after the experiment, I expect this calibration to be done. These are mentioned p16 l22. Along the same lines, a list of future QC to be applied is advocated p15. I would be reluctant to use such a data set. My conception of publishing a data set in such a journal is that final QC should be performed beforehand, and future users should not worry about it.

- Section 2.2.2: It is never specified that the gliders were set to surface every 3 (deep) and 10 (shallow) dives. Estimates of depth-average currents by gliders between consecutive surfacing should be mentioned. Those are essential to infer geostrophic velocities. The sampling strategy unfortunately divides by 3 and 10 the number of current estimations. What was the aim of this sampling strategy? Moreover, when the glider does not spend equally distributed time at each depth level, depth-average currents can not be treated as such anymore. How does the QC deal with this issue? To my mind, this is a real weakness of the glider dataset, especially in an experiment dedicated to submesoscale. I discovered this point by looking at the glider data. Readers should be made aware of this in the manuscript.

- Section 3.3.2: How in-house QC differ from international standard for profiling floats and gliders?

Specific comments:

p2 l32 "thanks due" p6 l2: Specify the glider type and sensors. p10 l1: wrong degree symbol, please also correct other instances.

[Figure]

2018.

---

## Author Comment (AC1) · 24 Dec 2018

First - Is this a "review" article? ESSD defines review articles as: ". . . may compare methods or relative merits of data sets, the fitness of individual methods or data sets for specific purposes, or how combinations might be used as more complex methods or reference data collections." As I read it from the manuscript this is not the case. The current version of the manuscript reads more as a copy of data information from individual reports and the data section in scientific publications related to the experiment. As it stands, I do not see the criteria for a "review" type article fulfilled.

We acknowledge the reviewer's comment concerning the nature of the article. We made a mistake during the submission process. Referring to the ESSD web page, we read that "Articles in the data section may pertain to the planning, instrumentation, and execution of experiments or collection of data.", and this is indeed the objective we had when submitting the manuscript. However in the Submission page, the "Manuscript Type" did not offer the possibility to select it, hence we took another one which seemed the closest. We have contacted the editorial office concerning this and the manuscript type was changed on September 18, 2018.

Significance

Three sub-criteria to evaluate:

• Uniqueness: It should not be possible to replicate the experiment or observation on a routine basis. Thus, any data set on a variable supposed or suspected to reflect changes in the Earth system deserves to be considered unique. This is also the case for cost-intensive data sets which will not be replicated due to financial reasons. A new or improved method should not be trivial or obvious. The data set is unique. (rating: 1 Excellent)

Thank you for the appreciation

• Usefulness: It should be plausible that the data, alone or in combination with other data sets, can be used in future interpretations, for the comparison to model output or to verify

other experiments or observations. Other possible uses mentioned by the authors will be considered.

The current manuscript does not provide information that promote the reuse of the data set (it may for subsets). No attempt is made to provide a structured overview about the workflow that is linked to the creation of the data set and, equally important, the QA/QC are not provided in a transparent way. For example, in the netcdf data files I see different QC flags provided – one is for example "SOCIB Quality control Data Protocol". What does that mean? This is not an international standard. A data set description, as envisioned in this ESSD submission, should exactly describe such non-standard QC procedures. Which QA and QC methods were applied (give brief description, DOIs if applicable)?

We agree with the reviewer and to address these issues:

- A new section dedicated to data reuse has been be added (see below) and

- the section "*3.3.2 Quality control*" has been expanded and made more explicit.
* * *
Added text:

**Data Reuse**

Three main types of data reuse are foreseen: 1. model validation, 2. data assimilation (DA) and 3. planning of similar in situ experiments.

With the increase of spatial resolution in operational models, the validation at the smaller scales requires high-resolution observations. Remote-sensing measurements such as SST or chlorophyll-a concentration provides a valuable source of information but are limited to the surface layer. In the case of the present experiment, the position, intensity (gradients) and vertical structure of the front represent challenging features for numerical models, even when data assimilation is applied (Hernandez-Lasheras and Mourre, 2018)).

The AlborEx dataset can be used for DA experiments, for example assimilating the CTD measurements in the model and using the glider measurements as an independent observation dataset. The assimilation of glider observations has already been performed in different regions (e.g. Melet et al., 2012; Mourre and Chiggiato, 2014; Pan et al., 2014) and has been shown to improve the forecast skills. However the assimilation of high-resolution data is not trivial: the the background error covariances tends to smooth the small scale features present in the observations.

Finally, other observing and modeling programs in the Mediterranean Sea can also benefit from the present dataset, for instance the Coherent Lagrangian Pathways from the Surface Ocean to Interior (CALYPSO) in the Southwest Mediterranean Sea (Johnston et al., 2018). Similarly to AlborEx, CALYPSO strives to study a strong ocean front front and the vertical exchanges taking place in the area of interest (see `https://www.onr.navy.mil/Science-Technology/Departments/Code-32/All-Programs/Atmosphere-Research-322/Physical-Oceanography/CALYPSO-DRI` for details).
* * *
We also complement the introduction with references to other studies using multi-platform approaches in the same area.

Added text:
Similar studies comparing almost synchronous glider and SARAL/AltiKa altimetric data on selected tracks have also been carried between the Balearic Islands and the Algerian coasts (Aulicino et al., 2018; Cotroneo et al., 2016).

I also miss any information how/if this data is disseminated via international data centres and how the data QC and dissemination is coordinate with the respective observing networks (Argo, DBCP, . . . ). Seadatanet is been mentioned in the text but it is unclear which specific recommendations are given.

(rating: 4 poor)

All the data presented in this paper are open data and can be accessed through the SOCIB Data Center in a few clicks, without any registration. Moreover, the data API (`http://api.socib.es`) strongly improves the data access to users and the dissemination to national or international data centers, which can easily establish a data transfer if they want to include SOCIB data into their portal.

As of today, many international databases exist and frequently, new ones are created with new projects, making the data landscape complex and the making it tedious to extensively document the data flow between SOCIB data and those databases. For instance:

- all the drifters data are transmitted to the Mediterranean Surface Velocity Programme (MedSVP, `http://doga.ogs.trieste.it/sire/medsvp/`);

- Most of the data are transmitted to the Mediterranean Operational Network for the Global Ocean Observing System (MONGOOS, `http://www.mongoos.eu/data-center`);

- MONGOOS sends the data to the In Situ Thematic Assembly Center (INSTAC) of the Copernicus Marine Environment Monitoring Service (CMEMS, `http://www.marineinsitu.eu`);

- The PROVBIO float is available in OAO database (Villefranche-sur-mer, `http://www.oao.obs-vlfr.fr/maps/en/`

- The Argo floats and drifters data are transmitted to the CMEMS INSTAC.

- . . .

Our approach to guarantee that the data are available to the widest community consists of

1. Having the data easily accessible in a standard format (netCDF) through standard protocols (HTTP, OPEnDAP, . . . ), and without any registration. This means that any user or entity can download all the files and include them in their portal or database.

2. Providing an efficient data API to make easier the data discovery: the role of the API is really to allow users to make request such as:

   - "provide me all the observations measured by the platform X (glider, drifter)" or

   - "provide me all the observations in the region located in the area Y during a given time period."

The explicit mention to SeaDataNet is made because of their Regional Data Products, which we believe are of crucial importance for the scientific community needing a complete set of historical, in situ data. The data transfer from SOCIB to SeaDataNet is foreseen in the future.

• Completeness: A data set or collection must not be split intentionally, for example, to increase the possible number of publications. It should contain all data that can be reviewed without unnecessary increase of workload and can be reused in another context by a reader.

It is difficult to evaluate this point. However, the nutrient data is not mentioned but is, according to Pascual et al. 2017 part of the AlborEX campaign. I would expect that these data set are described here as well (and respective QC (e.g. GO-SHIP nutrient manual??) and associated uncertainty estimates.

(rating: 2 to 3)

We agree with this suggestion and will add a specific section dedicated to the nutrient data. In relation to these data, we wish to add to the list of co-authors:

- Antonio Tovar-Sánchez, Instituto de Ciencias Marinas de Andalucía, (ICMAN – CSIC), Puerto Real, Spain and

- Eva Alou, SOCIB,

who were responsible for the acquisition and processing of these data during and after the cruise.

We have now included the dissolved inorganic nutrients measured during Alborex in the new file AlborexPerseus2014_LabSamplesNutrients_L1.nc, available at https://repository.socib.es:8643/repository/entry/show?entryid=07ebf505-bd27-4ae5-aa43-c4d1c85dd500. The files still has to be included to the general thredds directory of SOCIB.

This text was added to the new manuscript:

> **Added text:**
>
> Samples for nutrient analysis were collected in triplicate from CTD Niskin bottles and immediately frozen for subsequent analysis at the laboratory. Concentrations of dissolved nutrients (Nitrite: $NO_2^-$, Nitrate: $NO_3^-$ and Phosphate: $PO_4^{3-}$ were determined with an autoanalyzer (Alliance Futura) using colorimetric techniques (Grasshoff et al., 1983). The accuracy of the analysis was established using Coastal Seawater Reference Material for Nutrients (MOOS-1, NRCCNRC), resulting in recoveries of 97%, 95% and 100% for $NO_2^-$, $NO_3^-$ and $PO_4^{3-}$, respectively. Detection limits were $NO_2^-$:0.005 $\mu$M, $NO_3^-$: 0.1 $\mu$M and $PO_4^{3-}$: 0.1 $\mu$M.

Data quality

The data must be presented readily and accessible for inspection and analysis to make the reviewer's task possible. Even if a data set submitted is the first ever published (on a parameter, in a region, etc.), its claimed accuracy, the instrumentation employed, and methods of processing should reflect the "state of the art" or "best practices". Considering all conditions and influences presented in the article, these claims and factors must be mutually consistent. The reviewer will then apply his or her expert knowledge and operational experience in the specific field to perform tests (e.g. statistical tests) and cast judgement on whether the claimed findings and its factors – individually and as a whole – are plausible and do not contain detectable faults.

I touched on that already under "Usefulness". In the manuscript no transparent QC assessment is presented. What were the methods of processing (provide key steps, DOI at least). What were, including quantification of uncertainties and qualification via flags, the results of the QA/QC procedures? Which were the major shortcomings of the data acquisition process and what could be done better in the future? For example, has the drifter data included in the European E–SurfMar data base and also in the DBCP global drifter data sets? Have the recommendations (Best Practices, Protocols) from E–SurfMar / DBCP considered? It looks like no commonly agreed standard has been used for some parameters – as "SOCIB Quality control Data Protocol" suggest? (rating: 3)

The QC procedure is described in the document

> QUID_DCF_SOCIB-QC-procedures.pdf
>
> SOCIB Quality Control Procedures
> Data Center Facility
> September 2018
> DOI: doi:10.25704/q4zs-tspv

The procedure in based on the commonly agreed standards.

The article has been re-organised and for each type of platform, a description of the quality checks performed on the corresponding data has been added.

Which were the major shortcomings of the data acquisition process and what could be done better in the future?

Possibly the glider sampling strategy could be improve by increasing the relative frequency of surfacing, in order to have more information on the variables near the surface.

Presentation quality

Long articles are not expected. Regarding the style, the aim is to develop stereotypical wording so that unambiguous meaning can be expressed and understood without much effort. The article should express clearly what has been found, where, when, and how. The article text and references should contain all information necessary to evaluate all claims about the data set or collection, whether the claims are explicitly written down in the article, or implicit, through the data being published or their metadata. The authors should point to suitable software or services for simple visualization and analysis, keeping in mind that neither the reviewer nor the casual "reader" will install or pay for it.

mostly OK (given the limitation outlined in the previous points). It would be useful to include a brief introduction into the "design of the experiment. Visualisation tools are not given. (rating: 2-3)

A section "*Design of the experiment has been added*" in Section 2, after the "*General oceano-graphic context*" References to existing visualisations tools have been provided in a new section "*4.3 Data reading and visualisation*". It is worth mentioning here that a set of Python functions are provided to read, process and visualise the content of type of file.

Figure 1: Access to the deep glider data: the in-house viewers are listed in the bottom left corner.

Added text:

**Design of the experiment**

The deployment of in situ systems was based on the remote-sensing observations described in the previous Section. Two high-resolution grids were sampled with the research vessel, covering an approximative region of 40 km $\times$ 40 km. At each station, one CTD cast and water samples for chlorophyll concentrations and nutrients analysis were collected. The thermosalinograph observations were also used in order to assess the front position.
One deep glider and one coastal glider were deployed in the same area with the idea to have butterfly-like track across the front. These idealised trajectories turned out to be impossible considering the strong currents occurring in the region of interest at the time of the mission. The 25 drifters were released close to the frontal area with the objective to detect convergence and divergence zones. Their release locations were separated by a few kilometers.

Also, when accessing the data through the catalog (doi:10.25704/z5y2-qpye), users have access to different viewers (depending on the type of data), in one click, as shown in the figure below.

The following paragraph has been added:

> **Added text:**
>
> When accessing the data catalog, users are provided a list of in-house visualisation tools designed to offer quick visualisation of the file content. The visualisation tools depend on the type of data: *JWebChart* is used for time series; *Dapp* displays the trajectory of a moving platform on a map; the *profile-viewer* allows the user to select locations on the map and view the corresponding profiles.

**Specific comments**

P2/l.4: I do not agree with the statement: "a perfect observational system would consist in dense array of sensors present at many geographical locations, many depths and measuring almost continuously a wide range of parameters..." – this "generalization" is trivial and useless. From an observing design point of view a "perfect" observing system must follow a design that will record only the observations that are needed to analyse the problem. As such the perfect observational system always depends on motivation for the experiment (or the problem in more general words) - in some cases a "perfect observing system" may comprise only one single sensor at one single depth at different locations if this has been found a sufficient approach for solving the problem (e.g. estimating global warming through a global tomography array). Please reformulate the statement along those lines.

We agree that this formulation was not adequate and rephrased this part following this comment, as follows:

> **Added text:**
>
> To properly capture and understand these small-scale features, one cannot settle for only observations of temperature and salinity profiles acquired at different times and positions, but rather has to combine the information from diverse sensors and platforms acquiring data at different scales and at the same time, similarly to the approach described in Delaney and Barga (2009). This also follows the recommendation for the Marine Observatory in Crise et al. (2018), especially the co-localization and synopticity of observations and the multi-platform, adaptive sampling strategy. We will refer to this as multi-platform systems, by opposition to experiments articulated only around the observations made using a research vessel. Further details can be found in Tintoré et al. (2013).

**Additional references**

Aulicino, G., Cotroneo, Y., Ruiz, S., Sánchez Román, A., Pascual, A., Fusco, G., Tintoré, J., and Budillon, G.: Monitoring the Algerian Basin through glider observations, satellite altimetry and numerical simulations along a SARAL/AltiKa track, Journal of Marine Systems, 179, 55–71, doi:10.1016/j.jmarsys.2017.11.006, URL https://www.sciencedirect.com/science/article/pii/S0924796317302658, 2018.

Cotroneo, Y., Aulicino, G., Ruiz, S., Pascual, A., Budillon, G., Fusco, G., and Tintoré, J.: Glider and satellite high resolution monitoring of a mesoscale eddy in the algerian basin: Effects on the mixed layer depth and biochemistry, Journal of Marine Systems, 162, 73–88, doi:10.1016/j.jmarsys.2015.12.004, URL https://www.sciencedirect.com/science/article/pii/S0924796315002298, 2016.

Crise, A., Ribera d'Alcalà, M., Mariani, P., Petihakis, G., Robidart, J., Iudicone, D., Bach-mayer, R., and Malfatti, F.: A Conceptual Framework for Developing the Next Generation of Marine OBservatories (MOBs) for Science and Society, Frontiers in Marine Science, 5, 1–8, doi:10.3389/fmars.2018.00318, URL https://www.frontiersin.org/articles/10.3389/fmars.2018.00318/full, 2018.

Delaney, J. R. and Barga, R. S.: Observing the Oceans - A 2020 Vision for Ocean Science, pp. 27–38, Microsoft Research, URL https://www.microsoft.com/en-us/research/publication/observing-the-oceans-a-2020-vision-for-ocean-science/, 2009.

Grasshoff, K., Kremling, K., and (Eds), M. E., eds.: Methods of Seawater Analysis, , doi:10.1002/9783527613984, URL https://onlinelibrary.wiley.com/doi/book/10.1002/9783527613984, 1983.

Hernandez-Lasheras, J. and Mourre, B.: Dense CTD survey versus glider fleet sampling: comparing data assimilation performance in a regional ocean model west of Sardinia, Ocean Science, 14, 1069–1084, doi:10.5194/os-14-1069-2018, URL http://dx.doi.org/10.5194/os-14-1069-2018, 2018.

Johnston, T. M. S., Rudnick, D. L., Tintoré, J., and Wirth, N.: Coherent Lagrangian Pathways from the Surface Ocean to Interior (CALYPSO): Pilot Cruise report, Tech. rep., Scripps Institution of Oceanography (SIO), URL http://scrippsscholars.ucsd.edu/tmsjohnston/files/calypssocibcruisereport2018.pdf, last accessed: December 17, 2018, 2018.

Melet, A., Verron, J., and Brankart, J.-M.: Potential outcomes of glider data assimilation in the Solomon Sea: Control of the water mass properties and parameter estimation, Journal of Marine Systems, 94, 232–246, doi:10.1016/j.jmarsys.2011.12.003, URL http://dx.doi.org/10.1016/j.jmarsys.2011.12.003, 2012.

Mourre, B. and Chiggiato, J.: A comparison of the performance of the 3-D super-ensemble and an ensemble Kalman filter for short-range regional ocean prediction, Tellus A: Dynamic Meteorology and Oceanography, 66, 21 640, doi:10.3402/tellusa.v66.21640, URL http://dx.doi.org/10.3402/tellusa.v66.21640, 2014.

Pan, C., Zheng, L., Weisberg, R. H., Liu, Y., and Lembke, C. E.: Comparisons of different ensemble schemes for glider data assimilation on West Florida Shelf, Ocean Modelling, 81, 13–24, doi:10.1016/j.ocemod.2014.06.005, URL http://dx.doi.org/10.1016/j.ocemod.2014.06.005, 2014.

Tintoré, J., Vizoso, G., Casas, B., Heslop, E., Pascual, A., Orfila, A., Ruiz, S., Martínez-Ledesma, M., Torner, M., Cusí, S., and et al.: SOCIB: The Balearic Islands Coastal Ocean Observing and Forecasting System Responding to Science, Technology and Society Needs, Marine Technology Society Journal, 47, 101–117, doi:10.4031/mtsj.47.1.10, URL http://www.ingentaconnect.com/content/mts/mtsj/2013/00000047/00000001/art00010;jsessionid=2cbcvta0m97c.x-ic-live-02, 2013.

---

## Author Comment (AC2) · 24 Dec 2018

**Major comments**

My major concerns on the actual version of the paper are the following: 1. I think the text is not well organized. Some info on the data is find in Section 2 (AlborEX mission) and in Section 3.3 (Data Processing). This spreading of information makes the search for information through the paper difficult. I would bring Section 3.3. earlier in the paper and avoid to spread the information for each platform in different sections. Some specific comments below are related to this problem (e.g. mention of flags even before introducing them).

The Section 3.3. has been moved earlier in the text, in the Section 2, so that the reader is aware of the processing and Quality Control done of the data. The information is now provided in two subsections:

- *"2.4 Processing levels"*, which has been extended and made clearer following other comments

- *"2.5 Quality control"*, where the general procedure is made explicit.

2. The QC control is a weakness in this manuscript as it suggests that some QC is done, but it is not very clear on which data and how it is done. For some instruments, QC flags and their meaning are embedded in the files (e.g. float and drifters), but some doesn't (glider files). This inconsistency is not so much a problem to me as long as it is clearly stated in the paper which files contains QC flags. These quality flags should however be defined in the text. There are several mentions of "quality flags" in the text and figure caption, but little explanation is provided on these. Figure 12 has 9 quality flags that are not even described (although I see their meaning in drifters and float files). Where the QC is easy to reference

(e.g. "file generated with Socib glider toolbox vX.X", or "File QC done using Socib standard procedure following a procedure described in a certain paper", etc.), it should be mention in the netCDF file as well.

To address these comments:

1. A new table stating the meaning of the quality flag has been created (Table 2).

2. A subsection "QC tests" has been inserted at the end of Section 2 to explain the general procedure for the quality control.

3. In Section 3, for each platform type, a description of the specificities of the QC has been appended.

Concerning the glider data: the toolbox referenced in the manuscript does not apply quality checks on the data in its current version. QC have been implemented but are still in testing phase. Once they are validated, the files will be reprocessed and made available.

More generally, a lot of efforts have been made to ensure that the provided data are of the highest quality, even if that was not reflected in the submitted manuscript. All the SOCIB quality checks are explicitly described in the following document:

QUID_DCF_SOCIB-QC-procedures.pdf

SOCIB Quality Control Procedures
Data Center Facility
September 2018
DOI: doi:10.25704/q4zs-tspv

and more tests are progressively developed in the current battery.

3. Why all processing level are not provided? The text suggests that all levels are provided (e.g. Table 3), but at the moment mostly L1 is provided. For gliders, L1 and L2 are provided. For the Float, L1 is provided for Arvor-A3 and Provor-Bio, but L0 for Arvor-C. Why? No explanation for this is provided (I think float data should be provided in L1 and L2 level as well). If some QC is applied on L1, maybe L0 should be provided as well to the future user? For glider L2 data, a choice is made regarding the vertical binning of the profiles. Which size these vertical bins are? This information should be provided somewhere.

Following the definitions adopted at the SOCIB data center, Level 2 only exists for glider measurements: it means that we go from 3-dimensional trajectories to a time series of profiles (the observations are spatially interpolated. The description of the processing levels has been edited and clarified in the new manuscript.

Missing L1 for Arvor-C: this comes from an oversight: the file has been made available in the new version of the dataset. The link to the thredds catalog is: http://thredds.socib.es/ thredds/catalog/drifter/profiler_drifter/profiler_drifter_arvorc001-ime_arvorc001/L1/2014/ catalog.html?dataset=drifter/profiler_drifter/profiler_drifter_arvorc001-ime_arvorc001/ L1/2014/dep0001_profiler-drifter-arvorc001_ime-arvorc001_L1_2014-05-25.nc

For the glider data gridding (from L1 to L2): the referee is correct, this has to be explained in the manuscript.

The gridding is performed by the function `gridGliderData` (https://github.com/socib/glider_toolbox/blob/master/m/processing_tools/gridGliderData.m), designed to get the glider trajectory data over instantaneous homogeneous regular profiles. By default, the vertical resolution (or step) is set to 1 meter in the present version of the processing, though it can be adapted by the user. For the spatial and temporal coordinates: they are computed as the mean values of the cast readings. For the variables: a binned is performed, taking the mean values of readings in depth intervals centered at selected depth levels.

These explanations are not in the new manuscript in the Section dedicated to the Processing levels.

4. Nowhere the sensor configurations are specified. I think a table gathering this information is worth it. For each platform, the list of sensor should be presented with their configuration (sampling frequency, ADCP ping-per-ensemble, ADCP vertical bin size, etc.). This should include all variables collected, for example, from the ship meteo station from which little information (or none) is present in the text. Same for the glider where there is Chl-a and turbidity data in the files, but these were not mentioned in the text. A table gathering this information would be useful.

We agree with the suggestion and provided this information in the manuscript. Instead of a table, we feel it is better to have the information distributed in each subsection referring to the different platforms. The manuscript has been modified accordingly.

5. A table regrouping all the platform with their basic configuration as well as their number of casts (when it applies) should be provided (sort of extended Table 3).

For each platform, we indicated the basic configuration as well as the number of casts (for CTD, gliders and Argo floats).

**Text-specific comments**

- Figure 1 too small (should take page width) - Figure 2 too small (should take page width)

Figures 1 and 2 have been enlarged in the new manuscript

- Figure 2 caption: there is mention of "flag data equal to 1" while these flag are not introduced in the text.

SST is not part of the dataset, we just use them to illustrate the situation during the mission, this is why we did not go into details concerning the flag = 1, which is explicitly described in the caption (good data).

- p.7, L1: The "total number of valid measurement" is not very useful. I would rather put the number of valid casts (see comment above on a new table with this info).

We agree. The number of valid measurements (for the gliders) has been removed and replaced by the number of casts, in the new manuscript.

- p.7, L6: "a spatial interpolation is applied on the original data, leading to the so-called Level-2 data, further described in Sec. 3.3." What does 'spatial interpolation' means? Section 3.3 is not very explicit on this. I know you mean that the glider yos have been separated into downward and upward casts and then assigned to a geographical coordinate, but maybe this should be stated explicitly (and I don't think "spatial interpolation" is an accurate description). Moreover, Is there any vertical interpolation done? Because there are still some NaNs in L2

data.

The referee is right, it is not exactly an interpolation that is performed, but a spatial gridding. The gridding is performed by the function `gridGliderData`, designed to get the glider trajectory data over instantaneous homogeneous regular profiles. By default, the vertical resolution (or step) is set to 1 meter. For the spatial and temporal coordinates: they are computed as the mean values among cast readings. For the variables: a binned is performed, taking the mean values of readings in depth intervals centered at selected depth levels. The NaN are indeed not removed by the binning process, but will be discarded or flagged once the file are re-processed with the new version of the Glider Toolbox.

This has been amended in the new manuscript, in the section that describes the different processing levels.

> **Added text:**
>
> **Level 2 (L2)** : this level is only available for the gliders. It consists of regular, homogeneous and instantaneous profiles obtained by gridding the L1 data. In other words, 3-dimensional trajectories are transformed into a set of instantaneous, homogeneous, regular profiles. For the spatial and temporal coordinates: the new coordinates of the profiles are computed as the mean values of the cast readings. For the variables: a binning is performed, taking the mean values of readings in depth intervals centered at selected depth levels. By default, the vertical resolution (or bin size) is set to 1 meter. This level was created mostly for visualization purposes.

- p.7, L15: "Interestingly, all the drifters exhibit a trajectory close to the front position" → Not clear what "trajectory close to the front means". Moreover, is that really surprising that surface drifter would aggregate on a front?

We remove the "Interestingly", as indeed it is expected and rephrased it to: "All the drifters moved along the front position (deduced from the SST images), until they encounter the Algerian Current".

- Figure 8 caption: "for the duration of the mission" → You mean the ship mission? Or the AlborEX campaign?

We meant for the AlborEX mission; this has been made explicit in the new manuscript. The caption now reads:

> **Added text:**
>
> Surface drifter trajectories. For the sake of simplicity and clarity, the temperature, when available, is only shown for the duration of the AlborEx mission (May 25-31, 2014)

- Figure 10: plots on the right column are of little information here (too low resolution to mean something), I would remove.

We agree that the resolution is not as good as the Arvor-C float, but for completeness we would prefer not to discard them.

- Table 1: "Period" should be replaced by "cycle length" as referred to in the text (Section 2.2.4).

Modified as suggested.

- Table 1: netCDF file for Provor-bio indicates deployment end date 2015-04- 24T12:02:59+00:00,

which is different from this table.

The correct date is indeed 2015-04-24T12:02:59+00:00. The table has been modified accordingly.

- Figure 11 caption: "quality flag" not defined.

Quality flag with a value of 1 (meaning "good data") is specified in the caption. We added a complete description in the text concerning this part.

- Section 3.3.1: A Section on processing levels, but they are not all provided. Why? I think all levels should be provided. This is related to a previous comment.

The origin of the initial decision of not providing the L0 data for all the files is twofold: For some platforms (gliders), the L0 files are rather large and contain many variables related to the platform engineering, no to oceanography. Even if the files were not provided through the Zenodo platform, they are still publicly available using the SOCIB thredds server. In the new version of the manuscript, we adopted a new way to distribute the data (the data catalog), in which the data files corresponding to all the processing levels are made available.

- p.14, Level 2 (L2): "obtained by interpolating the L1 data" → How L2 is obtained by "interpolating" L1? Isn't L1 cut into casts that makes L2?

Correct. It is not an interpolating but a gridding. The explanation of how this gridding is performed has been added to the manuscript.

- p.14, Level 2 (L2): "It is only provided for gliders, mostly for visualization and post-processing purposes: specific tools designed to read and display profiler data can then be used the same way for gliders." → Is there a problem with this sentence? I don't understand it.

We removed the part of the sentence starting with "post-processing purposes"

- Section 3.3.1 / Table 3: Is L1 level for float equivalent to L2 level for glider? For consistency, I think profiling float should have L1 and L2 data as well since these instruments have similarities on the way they profile the water column...

The L1 glider data consists of a 3-dimensional trajectories, which means that both the longitude, latitude and depth change with respect to time. The Level 2 aims to have the same data on vertical profiles: the longitude and latitude don't change for a given profile. This is illustrated in the figure below.

- p.12, L1: "This type of current measurements requires a careful processing in order to get meaningful velocities from the raw signal" → Why? What are the limitations that makes this instrument more sensitive compare to other ones?

The main reason for this sensitivity is the fact that the vessel's velocity is one or two order or magnitudes greater than the currents that have to be measured. It is thus critical to have good measurements of the vessel heading and velocity.

A sentence has been inserted at the beginning of that paragraph and we removed the sentence "*hence it is relevant to have a quality flag (QF) assigned to each measurement*".

- p.12, L4: "Figure 12 shows the QF during the whole mission." → How QF are calculated?

The QC procedure for the VM-ADCP is complex as it involves tests on a large number of variables such as:

    Bottom Track Direction

[Figure]

Bottom Track Velocity

Bottom Track error on velocity

Bottom Track Depth from beam

Sea water noise amplitude

. . .

with dependencies between them but also variables related to the vessel position and behavior (pitch, roll, speed, . . . ). The tests adopted are listed in the reference QUID document:

QUID_DCF_SOCIB-QC-procedures.pdf

SOCIB Quality Control Procedures
Data Center Facility
September 2018
DOI: doi:10.25704/q4zs-tspv

and the new manuscript now contains a summary of the ADCP QC procedure.

> **Added text:**
>
> The vessel's velocity is one or two order or magnitudes greater than the currents that have to be measured, hence this type of current measurements requires a careful processing in order to get meaningful velocities from the raw signal. The QC procedure for the VM-ADCP is complex as it involves tests on more than 40 technical and geophysical variables (SOCIB Data Center, 2018). The different tests are based on the technical reports of Cowley et al. (2009) and Bender and DiMarco (2009), which aim primarily at ADCP mounted on moorings. The procedure can be summarised as follows:
>
> 1. Technical variables: valid ranges are checked for each of these variables: if the measurement is outside the range, the QF is set to 4 (bad data). Example of technical variables are: bottom track depth, sea water noise amplitude, correlation magnitude.
>
> 2. Vessel behaviour: its pitch, roll and and orientation angles are checked and QF are assigned based on specific ranges. In addition the vessel velocity is checked and anomalously high values are also flagged as bad.
>
> 3. Velocities: valid ranges are provided for the computed current velocities: up to 2 m/s, velocities considered as good; between 2 and 3 m/s, probably good, and above 3 m/s, bad.

- Figure 12: Too small.

the figure has been enlarged in the new manuscript.

- Figure 12 and text below: 9 different quality flag are presented without any introduction on how they are calculated. The new paragraph in the same section (see comment before) now explains how the quality flag are assigned.

- Section 3.3.2 is very short. Should be re-worked following comments above. We agree that the section dedicated to the Quality Control was too short. The QC are now described as follows: A general description in Section "2.5.2 QC tests" and Specific explanations of the tests performed for each platform, making that part more self-contained.

**Comments on data files**

The dataset consists of a relatively large number of files. I did my best but it was nearly impossible to review them all in details.

We really appreciate your time to extensively check of the files.

Here are some comments: - There are very large spikes in deep glider turbidity

yes, as the provided datasets for gliders have not undergone the quality checks (yet), there are still spikes and bad values for some of the variables. The text has been modified accordingly.

- There are missing data for about 10h in deep glider data between May 25-26. Unless I missed it, no explanation for this are provided.

The referee is right, some data are missing because the glider payload suffered an issue with the data logging software, resulting in no data acquisition during a few hours, during which the problem was being fixed. After that the data acquisition could be resumed.

This explanation has been added to the corresponding section in the new manuscript.

[Figure]

Added text:

On May 25 at 19:24 (UTC), the deep glider payload suffered an issue with the data logging software, resulting in no data acquisition during a few hours, during which the problem was being fixed. After this event, the data acquisition could be resumed on May 26 at 08:50 (UTC).

- Oxygen data for both glider seems to suffer from thermal lag problems

Yes it is true, we have reached the same conclusion when checking the oxygen data. The issue comes from the sensitivity of the optode to the temperature and the time response of the temperature sensor.

Comparing the temperature obtained with the glider CTD and the temperature of the oxygen sensor (next Figure) also highlights the lag existing between the 2.

To the extent of our knowledge, there is not yet an agreement from the community on how to correct this lag. Nicholson and Feen (2017) proposed a calibration based on the measurements made with the glider optode of the oxygen partial pressure of the atmosphere. Such a procedure can be contemplated in the near future.

Added text:

Finally, oxygen concentration measurements (not shown here) seem to exhibit a lag. According to Bittig et al. (2014), this issue is also related to the time response of oxygen optodes. As far as we know, there is not yet an agreement from the community on how to correct this lag, this is why the data are kept as they are in the present version, though we don't discard an improvement of the glider toolbox to address this specific issue.

- Provor-bio datafile contains levels down to over 7000 m. Some problems are found: 1. Why such long level dimension?

The 7000 comes is the depth dimension, as shown by the "ncdump -h" output:

```
dimensions:
    time = UNLIMITED ; // (71 currently)
    depth = 7118 ;
    name_strlen = 49 ;
```

But it does not mean that the maximal depth is actually 7000 m or deeper, as it depends on the vertical resolution. Here the deepest measurements are on the order of 1000 m. The profiles from PROVBIO are shown in the next 2 figures.

2. No good data is found below 3̃25m, although Table 1 suggest that the float is profiling to 1000m

We confirm that the float acquired data up to approx. 2000 m, even though the vertical resolution is not as high as near the surface. We reproduce (see below) the Figure 10 from the manuscript, this time without limiting the depth range, in order to confirm the availability of data at that depth.

- Arvor A3 data file suffers from similar problem: file contains data only down to 115m while Table 1 says 2000m

For the Arvor A3 we confirm that profiles are available up to approx. 2000 m. The "115" mentioned above are in fact the number of vertical levels provided in the file, not the final depth. Also see the figure above for the data availability.

- Arvor-C data file (only L0 provided) do not contain metadata (no file attributes, etc.). In addition, missing data (at least for temperature) appears to me as very large numbers (9.969210e+36) that makes them difficult to manipulate.

The L0 file with the metadata and the L1 file have been prepared and are now available. The link to the thredds catalog are provided below: L0: http://thredds.socib.es/thredds/catalog/drifter/profiler_drifter/profiler_drifter_arvorc001-ime_arvorc001/L0/2014/catalog.html?dataset=drifter/profiler_drifter/profiler_drifter_arvorc001-ime_arvorc001/L0/2014/dep0001_profiler-drifter-arvorc001_ime-arvorc001_L0_2014-05-25.nc L1: http://thredds.socib.es/thredds/catalog/drifter/profiler_drifter/profiler_drifter_arvorc001-ime_arvorc001/L1/2014/catalog.html?dataset=drifter/profiler_drifter/profiler_drifter_arvorc001-ime_arvorc001/L1/2014/dep0001_profiler-drifter-arvorc001_ime-arvorc001_L1_2014-05-25.nc

R/V Socib CTD and thermosalinograph files say that units of temperature are "C". I prefer the convention from glider files which uses "Celsius".

We take note of the suggestion and will perform the modification in a new release of the data files, as it involves a re-processing of several files from other missions). The referee is totally right, as the Unidata documentation (https://www.unidata.ucar.edu/software/netcdf/netcdf/Units.html) states that "Celsius" should be used, "C" meaning "Coulomb".

**Minor comments**

- p.2; L23: "makes it possible" → makes possible

[Figure]

Figure 1: Salinity profiles acquired by the PROVBIO float. The 2nd panel depicts the profile in the 500–1000 m layer.

[Figure]

Figure 2: Adapted figure 10 of the manuscript, with the maximal depth of the profiles displayed.

corrected

- p.2; L23: "creation and publication of aggregated datasets covering the Mediterranean Sea" → SeaDataNet is not only about the Mediterranean replaced by "covering different European regional seas, including the Mediterranean Sea"

- p.2; L32: "thanks due to" → thanks to

corrected (removed "due")

- Section 2.2.1: "CTD surveys" or CTD legs?

corrected (legs)

- Glider L1 files (e.g. `dep0012_ideep00_ime-sldeep000_L1_2014-05-25_data_dt.nc`) say that the project is "PERSEUS". Is that right? There is no mention of the AlborEX project in the file header.

Correct, AlborEx was the Subtask 3.3.4 of PERSEUS project, but in this case AlborEx was not explicitly mentioned in the file header. This will be added during the next re-processing of the data files.

- p.10, L1: problems with latitude longitude degree symbol.

corrected

- p.10, L5: temperature, salinity and T,S is use on the same line. Please homogenize.

replaced by "In addition to these variables"

- p.12, L17: "Network Common Data Form (netCDF, `https://doi.org/(http://doi.org/10.5065/D6H70CW6`, last accessed on August) 3, 2018)" Is there a mis-placed parenthesis?

Corrected, the "(" after `.org` has been removed.

- p.13, L2: problem with file name (too long for page)

Corrected (new line added).

- p.16, L25: How stable in time the python codes made available on Github will be?

Generally, reading netCDF files with Python is an easy task, as it is with other languages (MATLAB, Julia, R), so we do not expect any difficulties for the data users. Here what we did is to provide a set of the Python codes written to show how to read the data and reproduce the plots of the papers, as we think it might save time if somebody wants to create something similar, or even reproduce the paper plot.

With Python it is relatively straightforward to use virtual environment, which allows one to work with specific version python modules. If a user works with a virtual environment which has the same packages versions as those specified on GitHub (file `requirements.txt`), then the code will run (since the netCDF files will be the same).

Even if issues occur, we think that providing the codes employed to manipulate the data files, along with the data, is a step toward the reproducibility of the results.

**Additional references**

Bender, L. and DiMarco, S.: Quality Control and Analysis of Acoustic Doppler Current Profiler Data Collected on Offshore Platforms of the Gulf of Mexico, Tech. rep., U.S. Dept. of the Interior, Minerals Mgmt. Service, Gulf of Mexico OCS Region, New Orleans, LA, 66 pp., 2009.

Bittig, H. C., Fiedler, B., Scholz, R., Krahmann, G., and Körtzinger, A.: Time response of oxygen optodes on profiling platforms and its dependence on flow speed and temperature, Limnology and Oceanography: Methods, 12, 617–636, doi:10.4319/lom.2014.12.617, URL `https://aslopubs.onlinelibrary.wiley.com/doi/abs/10.4319/lom.2014.12.617`, 2014.

Cowley, R., Heaney, B., Wijffels, S., Pender, L., Sprintall, J., Kawamoto, S., and Molcard, R.: INSTANT Sunda Data Report Description and Quality Control, Tech. rep., CSIRO, 2009.

Nicholson, D. P. and Feen, M. L.: Air calibration of an oxygen optode on an underwater glider, Limnology and Oceanography: Methods, 15, 495–502, doi:10.1002/lom3.10177, URL `https://aslopubs.onlinelibrary.wiley.com/doi/abs/10.1002/lom3.10177`, 2017.

SOCIB Data Center: SOCIB Quality Control Procedures, Tech. rep., Balearic Islands Coastal Observing and Forecasting System, Palma de Mallorca, Spain, doi:10.25704/q4zs-tspv, URL `http://repository.socib.es/repository/entry/show?entryid=a85d659d-b469-4340-ae88-c361333c68b6`, 2018.

---

## Author Comment (AC3) · 24 Dec 2018

The instrument specifications have been added in the manuscript: for each platform, a subsection "*Configuration*", containing the information about the platform and variables, has been added.

- Were they any water sample taken during the cruise in order to calibrate the CTD, or chlorophyll-a fluorescence? More than four years after the experiment, I expect this calibration to be done. These are mentioned p16 l22. Along the same lines, a list of future QC to be applied is advocated p15. I would be reluctant to use such a data set. My conception of publishing a data set in such a journal is that final QC should be performed beforehand, and future users should not worry about it.

You are right, water samples were collected.

The CTD data calibrated using the bottle data are available as a new processing level called `L1_corr`, and now described in the manuscript. Concerning the chlorophyll-a fluorescence calibration: it is correct that the calibration has not yet been performed. The decision to publish the data in the present state comes from a balance between:

- The will to share as soon as possible that dataset with the research community interested in the submesoscale, knowing that articles using the dataset have already been published.

- The need to have the best quality for the dataset.

Even if there may still be room for improvement in terms of quality control, for instance by creating new quality checks, our conviction is that the dataset in its current state is mature enough to be employed by other researchers

- Section 2.2.2: It is never specified that the gliders were set to surface every 3 (deep) and 10 (shallow) dives. Estimates of depth-average currents by gliders between consecutive surfacing should be mentioned. Those are essential to infer geostrophic velocities. The sampling strategy unfortunately divides by 3 and 10 the number of current estimations. What was the aim of this sampling strategy? Moreover, when the glider does not spend equally distributed time at each depth level, depth-average currents can not be treated as such anymore. How

does the QC deal with this issue? To my mind, this is a real weakness of the glider dataset, especially in an experiment dedicated to submesoscale. I discovered this point by looking at the glider data. Readers should be made aware of this in the manuscript.

Thanks for mentioning this issue. It is indeed something that was not properly addressed in the initial manuscript.

We also believe that it is essential

- to have measurements near the surface to tackle oceanic processes and

- the highest frequency of profiles near the surface in order to properly estimate the depth-integrated velocity.

The reason why the gliders did not go to the surface for every profile arises from safety concerns: the intense marine traffic (see for example the density maps of MarineTraffic) and the existence of a Traffic Separation Scheme (TSS) near the sampling area were taken into account for the decision to limit the glider surfacing.

We added a paragraph in the subsection "*Configuration*" with the "*Gliders*" section:

> **Added text:**
> Due to safety concerns, both the deep and coastal gliders had their surfacing limited: the deep glider came to the surface one in every 3 profiles, while the coastal gliders came out one in every 10 profiles. While this strategy does not appear optimal in a scientific point of view (loss of measurements near the surface, meaning of the depth-average currents), the priority was set on the glider integrity.

- Section 3.3.2: How in-house QC differ from international standard for profiling floats and gliders?

In-house quality control are in fact based on international standards. The idea is not to reinvent the wheel but to use what already exists and add other contributions whenever possible. All the QC are detailed in:

> QUID_DCF_SOCIB-QC-procedures.pdf
>
> SOCIB Quality Control Procedures
> Data Center Facility
> September 2018
> DOI: doi:10.25704/q4zs-tspv

and the quality control is re-organised as follows:

1. A general section explaining the approach for the quality control.

2. For each platform, a sub-section describing the specificities in terms of QC.

As all the procedures are explained in the aforementioned document, for the sake of conciseness, we prefer to keep a summarised version in the manuscript.

**Specific comments**

p2 l32 "thanks due"

corrected

p6 l2: Specify the glider type and sensors

**Coastal:** Teledyne Webb Research Corp. Slocum, 1st generation, shallow version (200 m)

**Deep:** Teledyne Webb Research Corp., Slocum, 1st generation, deep version (1000 m)

This information is now included in Table 3 in the subsection "*3.2.1 Configuration*" related to the Gliders, along with the sensors and other technical data.

Overall, the descriptions of all the instruments and sensors have been extended and improved.

p10 l1: wrong degree symbol, please also correct other instances.

Corrected